# MagR: Weight Magnitude Reduction for Enhancing Post-Training Quantization

**Aozhong Zhang**[1]  **Naigang Wang**[2]  **Yanxia Deng**[1]  **Xin Li**[1]  **Zi Yang**[1]  **Penghang Yin**[1]

[1]University at Albany, SUNY    [2]IBM T. J. Watson Research Center

{azhang3, ydeng5, xli48, zyang8, pyin}@albany.edu

nwang@us.ibm.com

## Abstract

In this paper, we present a simple optimization-based preprocessing technique called Weight **Mag**nitude **R**eduction (MagR) to improve the performance of post-training quantization. For each linear layer, we adjust the pre-trained floating-point weights by solving a channel-wise $\ell_\infty$-regularized optimization problem. This process greatly diminishes the maximum magnitude of the weights and smooths out outliers, while preserving the layer's output. The preprocessed weights exhibit reduced range, which facilitates the subsequent quantization process. To implement MagR, we address the $\ell_\infty$-regularization by employing an efficient proximal gradient descent algorithm. Unlike existing preprocessing methods that involve linear transformations and subsequent post-processing steps, which can introduce significant overhead at inference time, MagR functions as a non-linear transformation, eliminating the need for any additional post-processing. This ensures that MagR introduces no overhead whatsoever during inference. Our experiments demonstrate that MagR achieves state-of-the-art performance on the Llama family of models. For example, we achieve a Wikitext2 perplexity of 5.95 on the LLaMA2-70B model for per-channel INT2 weight quantization without incurring any inference overhead. The code is available at https://github.com/AozhongZhang/MagR

## 1 Introduction

Large language models (LLMs) have achieved outstanding performance across a broad range of applications, demonstrating remarkable success. However, their unprecedented model size has led to many computation operations and substantial memory footprints, becoming significant barriers to their practical deployment and adoption in production environments. Accordingly, it is highly desirable to develop efficient model compression techniques for LLMs so they can be more widely deployed in resource-limited scenarios. Among the various techniques to compress and accelerate deep neural networks (DNNs), low-precision quantization has proven to be highly effective across numerous application domains and is widely adopted for accelerating DNNs. For LLMs, the inference runtime is dominated by the token generation process, where output tokens are produced sequentially, one at a time. This process is known to be memory bandwidth bound [3, 19]. As a result, the quantization of LLMs has primarily focused on reducing the bit-width of model weights, with the dual goals of lowering the model's footprint to enable deployment on resource-constrained devices and decreasing the memory bandwidth requirements to improve computational efficiency and accelerate inference.

The enormous computational demands for pre-training and fine-tuning Large Language Models (LLMs) have led to the emergence of Post-Training Quantization (PTQ) [4, 15, 22, 24, 27, 31, 41, 51, 52, 53, 43, 40], as a promising solution for quantizing these models. Unlike Quantization Aware Training (QAT) [7, 9, 12, 18, 21, 23, 46, 47, 48, 49], which is designed to minimize a global training loss for quantization parameters, PTQ directly applies low-precision calibration to a pre-

trained full-precision model using a minimal set of calibration samples. By aiming to identify an optimal quantized model locally through the minimization of a simplified surrogate loss, PTQ offers computational savings and resource efficiency compared to QAT. However, PTQ often lags behind QAT in accuracy, particularly for ultra-low precision lower than 4-bit. Thus, it remains an open problem to achieve an improved balance between cost and performance for PTQ-based approaches.

**Motivation.** To achieve state-of-the-art performance, the latest advances in PTQ [8, 25, 26, 36, 42] have proposed applying a linear transformation to process the pre-trained weights within a linear layer. This strategy of linear transformation aims to make the weights more suitable for the subsequent quantization procedure by reducing their magnitudes and suppressing outliers. In a nutshell, given the features $X$ and weights $W$, one constructs linear transformation $T$ such that $TW$ is better conditioned than $W$ in terms of being quantization-friendly. Such designs of $T$ include diagonal matrices (so-called channel-wise scaling) [25, 36, 42], random transformations [8, 39], and finite frames [1, 13]. Then, quantization is performed on $TW$ instead of the original weights $W$. To preserve the layer's output, however, the inverse transformation $T^{-1}$ has to be in turn applied to the features $X$, namely,

$$XW = (XT^{-1})(TW) \approx (XT^{-1})\mathcal{Q}(TW),$$

with $\mathcal{Q}(TW)$ being the quantized weights. PTQ done this way requires modifications on the original neural architecture, which involves additional computations of $XT^{-1}$ and extra memory storage for $T^{-1}$ at inference time. As a result, these steps introduce overhead that offsets the benefits provided by quantization. This raises a natural question:

*Can we effectively process the weights at the preprocessing stage to facilitate quantization without introducing inference overhead?*

To address this problem, we propose a simple optimization-based technique called Weight Magnitude Reduction (MagR). MagR functions as a non-linear transformation on weights without altering the original features/activations. The optimization program is designed to find new weights with minimal maximum magnitude, i.e., the $\ell_\infty$ norm, while preserving the layer's outputs.

**Contributions.** We propose a non-linear approach, MagR, based on channel-wise $\ell_\infty$-regularized least squares, to reduce the quantization scale without compromising the performance of pre-trained model, facilitating subsequent weight quantization while requiring no post-processing or inference overhead. See Figure 1 for comparing weight magnitudes before and after applying MagR. To address the $\ell_\infty$-regularization problem, we develop an efficient and parallelizable proximal gradient descent algorithm that involves computing $\ell_1$-ball projections at each iteration. Specifically, MagR preprocessing on a single Nvidia A100 GPU takes merely 15 min for LLaMA2-7B and 3.5 hr for the 70B model. Our results on INT weight-quantization demonstrate that MagR can significantly boost the performance in the sub-4bit regime when combined with fast gradient-free methods for layer-wise PTQ, such as rounding-to-nearest (RTN) [30] and OPTQ [16]. This approach achieves performance for weight quantization at least comparable to state-of-the-art PTQ methods on natural language processing (NLP) tasks, including gradient-based methods using block-wise reconstruction.

## 2   Related Work

Recently, as the sizes of language models are exploding, there has been growing interest in developing post-training quantization (PTQ) methods [8, 16, 25, 26, 36, 44, 45] for large-scale AI models like large language models (LLMs) to reduce the model sizes and accelerate inference by representing weight matrices in low precision. PTQ methods directly find the low-precision representation of the model without re-training, thereby preferred by extreme large-scale AI models. The OPTQ [16] uses approximate second-order information to calibrate the quantization. The method successfully compresses LLMs into 3 or 4 bits and can achieve reasonable accuracy in 2 bits. Researchers have found that the extreme values and the distribution of the weight entries highly affect the quantization errors and the quantized model quality. The original weight can be converted into a more quantization-friendly one by linear transformations. The approach can significantly reduce the quantization errors while bringing more time overhead during inference because of the linear transformation. OmniQuant [36] proposes learnable weight clippings and equivalent transformations to avoid the influence of extreme values. AWQ [25] searches for the most significant entries in the weight by looking at the activation and selects the scales that protect these entries. SmoothQuant [44] passes the difficulty

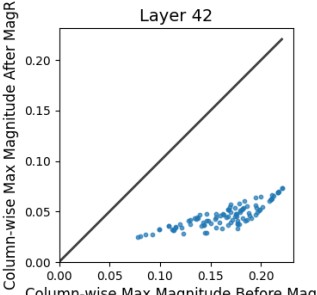 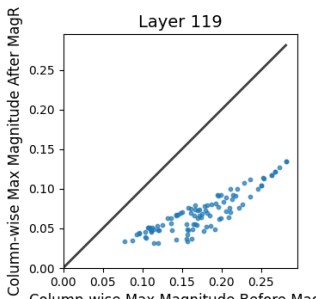 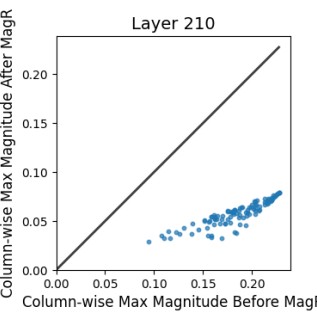

Figure 1: **Motivation behind MagR**: we can effectively reduce the magnitude of weights at the preprocessing stage. Each point denotes the maximum magnitude before ($x$-coordinate) and after ($y$-coordinate) applying MagR within a sampled channel (or column) of the weight matrix from three random layers of LLaMa2-7B [38]. These column-wise maximum magnitudes are typically more than halved through MagR.

in activation quantization to weights by an equivalent linear transformation. QuIP [8], AffineQuant [26] and FrameQuant [1] apply a linear transformation before quantization to make the transformed weight quantization-friendly. These approaches achieve high performance for extreme bits, like 2 bits, but introduce additional inference overhead though the transformation is carefully designed to be efficient. OmniQuant [36] and AffineQuant [26] can be adopted for weight-activation quantization by considering the activations in the proposed methods. The work [45] introduces a low-rank compensation method on top of other quantization methods, which employs low-rank matrices to reduce quantization errors with a minimal increase in model size. By modeling the quantization residual as an $\ell_\infty$-bounded perturbation, [2] proposes applying an $\ell_1$ penalty on the gradient of loss to enhance quantization robustness.

The works most closely related to ours are [20] and [27], both utilizing $\ell_\infty$ norm to regularize or constrain the weight range to a smaller scale. The Range Regularization ($R^2$) method [20] applies an $\ell_\infty$ penalty or its variants to the conventional network loss to regularize the weight range during end-to-end model pre-training, optimized via SGD. However, this approach becomes practically infeasible for large-scale models. In [27], a layer-wise pre-processing technique is proposed, which involves solving an intractable $\ell_0$-minimization problem while constraining the $\ell_\infty$-norm of weights.

## 3 Background

First, we clarify the mathematical notations that will be used throughout this paper:

**Notations.** We denote vectors by bold small letters and matrices by bold capital ones. For a positive integer $n$, $[n] := \{1, 2, \ldots, n\}$ denotes the set containing all positive integers up to $n$. For any two vectors $\boldsymbol{x}, \boldsymbol{y} \in \mathbb{R}^n$, $\langle \boldsymbol{x}, \boldsymbol{y} \rangle := \sum_{i=1}^n x_i y_i$ is the inner product. We denote by $\|\boldsymbol{x}\| := \sqrt{\langle \boldsymbol{x}, \boldsymbol{x} \rangle} = \sqrt{\sum_{i=1}^n x_i^2}$ the Euclidean norm; $\|\boldsymbol{x}\|_1 := \sum_{i=1}^n |x_i|$ is the $\ell_1$-norm; $\|\boldsymbol{x}\|_\infty := \max_{1 \le i \le n} |x_i|$ is the $\ell_\infty$-norm. For any matrix $\boldsymbol{X} \in \mathbb{R}^{m \times n}$, $\boldsymbol{X}^\top \in \mathbb{R}^{n \times m}$ is the transpose. We denote the spectrum norm of $\boldsymbol{X}$ by $\|\boldsymbol{X}\| = \sigma_{\max}(\boldsymbol{X})$, which equals its maximum singular value. Its Frobenius norm is given by $\|\boldsymbol{X}\|_F = \sqrt{\sum_{i=1}^m \sum_{j=1}^n X_{i,j}^2}$. Moreover, for vectors $\boldsymbol{x}$ and $\boldsymbol{y}$, $\boldsymbol{x} \odot \boldsymbol{y} := (x_1 y_1, \ldots, x_n y_n) \in \mathbb{R}^n$ denotes the Hadamard or element-wise product, and likewise for two matrices.

**Layerwise PTQ.** Post-training quantization via layerwise reconstruction calls for solving a least squares problem with a discrete constraint. For the pre-trained weights $\boldsymbol{W}$ within a linear layer, we aim to find the quantized weights $\boldsymbol{W}_q$ that minimize the following function

$$\min_{\boldsymbol{W}_q \in \mathbb{Q}} \|\boldsymbol{X} \boldsymbol{W}_q - \boldsymbol{X} \boldsymbol{W}\|_F^2, \tag{1}$$

where $\boldsymbol{X} \in \mathbb{R}^{(b \cdot l) \times m}$ is the feature matrix associated with a batch of calibration data consisting of $b$ samples stacked together, and each data sample is represented by an $l \times m$ sub-matrix. $\mathbb{Q} \subset \mathbb{R}^{m \times n}$ is an appropriate set of all feasible quantized weights.

The most straightforward PTQ technique, known as RTN, involves directly rounding the weight matrix $W$ without utilizing any additional data. An improvement over RTN was introduced by AWQ [25], which enhances the quantization process by incorporating channel-wise scaling on $W$.

Thanks to the simplicity of the layer-wise formulation (1), several efficient gradient-free algorithms [4, 16, 51, 53] have been recently proposed to address layer-wise quantization, including OPTQ. Built on top of OPTQ, QuIP subjects $X$ and $W$ to random orthogonal transformations to produce "incoherent" weight and Hessian matrices, leading to superior accuracy with sub-4bit quantization. However, this advantage comes with a trade-off; during inference, QuIP requires random orthogonal transformations on the feature inputs of linear layers, rendering noticeably slower throughput compared to OPTQ.

**Uniform Quantizer.** Given a set of points $w \in \mathbb{R}^m$, the commonly-used (asymmetric) uniform quantizer [9] defines the quantization step $\delta = \frac{\max(w) - \min(w)}{2^b - 1}$ and zero-point $z = \left\lfloor \frac{\min(w)}{\delta} \right\rceil$, and it quantizes $w$ onto the scaled integer grids $\mathbb{Q} = \{z \cdot \delta, (z+1) \cdot \delta, \ldots, (z + (2^b - 1)) \cdot \delta\}^m$ as follows:

$$w_q = \delta \cdot \left( \text{clamp}\left( \left\lfloor \frac{w}{\delta} \right\rceil - z, 0, 2^b - 1 \right) + z \right).$$

In per-channel (or per-group) PTQ, the quantization step $\delta$ is conventionally calculated based on the channel-wise (or group-wise, respectively) minimum and maximum values of the pre-trained weights $W$, as defined above, and remains constant throughout the quantization procedure.

## 4 The Proposed Method

In this section, we present the Weight Magnitude Reduction (MagR) method based on $\ell_\infty$-norm regularization, which is applied just before the quantization step within each linear layer. The intuition behind MagR is based on the following simple estimate of the layer-wise quantization error. Given the feature/activation matrix $X$, the quantizer $\mathcal{Q}$, and any pre-trained weights $w \in \mathbb{R}^m$, we have:

$$\min_{w_q \in \mathbb{Q}} \|Xw_q - Xw\| \leq \|X(\mathcal{Q}(w) - w)\| \leq \|X\|\|\mathcal{Q}(w) - w\| \leq \frac{\sigma_{\max}(X)\sqrt{m}}{2}\delta,$$

where $\delta = \frac{\max(w) - \min(w)}{2^b - 1}$ is the quantization step size. This shows that reducing the range of weights helps to suppress the quantization error. With this in mind, MagR preprocessing is designed to achieve two key effects:

- First, it effectively reduces the channel-wise (or column-wise) maximum magnitude of the weights, as illustrated by Figure 1.
- Second, it preserves the model's original performance with minimal accuracy loss after preprocessing. Table 1 demonstrates that MagR preprocessing maintains the perplexity of the pre-trained models, with only minor degradation.

Table 1: **A comparison of perplexity (PPL) for the original pre-trained and the MagR-processed LLaMA2 models.**

| Model | Method | Wikitext2 (PPL↓) | C4 (PPL↓) |
|---|---|---|---|
| LLaMA2-7B | Original | 5.47 | 6.97 |
| | MagR | 5.52 | 7.04 |
| LLaMA2-13B | Original | 4.88 | 6.46 |
| | MagR | 4.92 | 6.52 |
| LLaMA2-70B | Original | 3.31 | 5.52 |
| | MagR | 3.35 | 5.56 |

### 4.1 Approximately Rank-Deficient Feature Matrix

To illustrate the idea behind the proposed MagR method, let us consider a pre-trained weight vector $\hat{w} \in \mathbb{R}^m$ of a linear layer and the associated feature input matrix $X$. MagR leverages the fact that the feature matrix $X$ across all layers of LLMs is approximately rank-deficient. Specifically, if $X$ is exactly rank-deficient, the linear system modeling the layer's output, $Xw = X\hat{w}$ with variables $w$, generally has infinitely many solutions. That is, for any $\nu$ in the non-trivial kernel space of $X$, we have that $w = \hat{w} + \nu$ preserves the layer's output. Among all solutions, MagR aims to identify the weight vector $w$ with the smallest extreme value in magnitude.

Table 2: **The statistics of (approximate) fraction ranks in percentage (%)** of feature matrix $X$ across all layers of LLaMA models. All feature matrices are approximately rank-deficient with a fraction rank less than 100%. Some of them are highly low-rank with a fraction rank $\approx 1\%$.

| Model | Min | Max | Mean | 25% Percentile | 75% Percentile |
|---|---|---|---|---|---|
| LLaMA1-7B | 0.2 | 99.07 | 70.41 | 65.09 | 81.80 |
| LLaMA1-13B | 1.42 | 99.90 | 83.85 | 75.07 | 96.71 |
| LLaMA1-30B | 0.73 | 99.85 | 84.40 | 79.76 | 99.46 |
| LLaMA1-65B | 1.17 | 99.90 | 83.11 | 82.76 | 98.71 |
| LLaMA2-7B | 0.1 | 99.95 | 76.83 | 67.71 | 91.02 |
| LLaMA2-13B | 0.44 | 99.76 | 78.30 | 66.54 | 98.58 |
| LLaMA2-70B | 0.1 | 99.71 | 81.55 | 74.90 | 99.56 |

In [8], the authors empirically observed that the Hessian matrix $X^\top X$ is approximately low-rank across all layers in open pre-trained (OPT) models [54]. Here we examined the feature matrix of LLaMA models [37, 38]. Our approximate fraction rank of the feature matrix $X$ is defined as the fraction of singular values of $X$ such that $\sigma(X) > 0.01 \cdot \sigma_{\max}(X)$. Table 2 illustrates that all feature matrices extracted from LLaMA models are indeed rank-deficient according to this definition.

## 4.2 MagR via $\ell_\infty$-Regularization

Let us consider the quantization of a weight vector for simplicity. Given pre-trained weight vector $\hat{w}$, we would like to find a new set of weights $w$ with the smallest maximum magnitude, such that the layer output is preserved up to a small error $\varepsilon > 0$, i.e.,

$$\min_{w \in \mathbb{R}^m} \|w\|_\infty \quad \text{subject to} \quad \|Xw - X\hat{w}\| \le \varepsilon.$$

To efficiently implement MagR, we consider the following mathematically equivalent $\ell_\infty$-regularization problem instead:

$$\min_{w \in \mathbb{R}^m} \frac{1}{2}\|Xw - X\hat{w}\|^2 + \alpha\|w\|_\infty \tag{2}$$

where $\alpha > 0$ serves as the regularization parameter, balancing fidelity against the $\ell_\infty$ regularizer. To maintain the output of the layer, $\alpha$ should typically be set to a small value. Indeed, let $w^*$ be the minimizer of (2), we have that the $\ell_2$ error of the layer's output introduced by MagR is $O(\sqrt{\alpha})$:

$$\|Xw^* - X\hat{w}\| \le \sqrt{\|Xw^* - X\hat{w}\|^2 + 2\alpha\|w^*\|_\infty}$$
$$\le \sqrt{\|X\hat{w} - X\hat{w}\|^2 + 2\alpha\|\hat{w}\|_\infty} = \sqrt{2\alpha\|\hat{w}\|_\infty},$$

where $\|\hat{w}\|_\infty$ is a constant independent of $\alpha$, and the second inequality uses that $w^*$ is the minimizer.

**Proximal Gradient Descent.** Note that $\ell_\infty$-norm is a convex but non-differentiable function. In theory, the optimization problem (2) can be simply solved by a subgradient algorithm, but it is significantly slower than the more sophisticated proximal gradient algorithm which matches the convergence rate of standard gradient descent.

With the step size $\eta > 0$, proximal gradient descent [32] takes the following iteration:

$$w^{k+1} = \text{prox}_{\eta\alpha\|\cdot\|_\infty}\left(w^k - \eta \nabla_w \frac{1}{2}\|Xw - X\hat{w}\|^2\Big|_{w=w^k}\right)$$
$$= \text{prox}_{\eta\alpha\|\cdot\|_\infty}\left(w^k - \eta \cdot X^\top X(w^k - \hat{w})\right) \tag{3}$$

where $\text{prox}_{t\|\cdot\|_\infty}$ with the scalar $t > 0$ is the (scaled) proximal operator of $\ell_\infty$-norm function, defined as

$$\text{prox}_{t\|\cdot\|_\infty}(v) := \arg\min_{x \in \mathbb{R}^m} \frac{1}{2}\|x - v\|^2 + t\|x\|_\infty.$$

To ensure the convergence of (3), it is sufficient to choose the step size

$$\eta \le \frac{1}{\lambda_{\max}(X^\top X)},$$

where $\lambda_{\max}(\boldsymbol{X}^\top \boldsymbol{X})$ is the maximum eigenvalue of $\boldsymbol{X}^\top \boldsymbol{X}$.

**Proximal Operator of $\ell_\infty$-Norm.** It remains to determine the proximal operator of $\ell_\infty$-norm. It turns out we can compute it by leveraging the celebrated Moreau decomposition [29, 32]: for any $t > 0$,

$$\text{prox}_{t\|\cdot\|_\infty}(\boldsymbol{v}) = \boldsymbol{v} - t \cdot \text{proj}_{\|\cdot\|_1 \leq 1}\left(\frac{\boldsymbol{v}}{t}\right). \tag{4}$$

That is, computing the proximal operator of $\ell_\infty$ norm amounts to evaluating the projection onto $\ell_1$ ball, which is defined as

$$\text{proj}_{\|\cdot\|_1 \leq 1}(\boldsymbol{v}) := \arg\min_{\boldsymbol{x} \in \mathbb{R}^m} \|\boldsymbol{x} - \boldsymbol{v}\|^2 \quad \text{subject to} \quad \|\boldsymbol{x}\|_1 \leq 1.$$

Fortunately, computing projection onto the $\ell_1$ ball is an established task, and there are several efficient algorithms available. For example, see [11] and the references therein. Here we adopted a simple algorithm of $O(m \log m)$ time complexity as in [14], which supports parallelizable or vectorized implementation for the projections of a batch of weight vectors, i.e., a weight matrix, as will be described in the next subsection. The implementation mainly involves sorting and soft-thresholding [50]; see Algorithm 3 and its derivation in Appendix A.1 for the details.

**MagR for Weight Matrix.** In practical implementation of MagR, we preprocess the entire weight matrix $\boldsymbol{W} = [\boldsymbol{w}_1, \ldots, \boldsymbol{w}_n] \in \mathbb{R}^{m \times n}$ within each linear layer. For per-channel quantization (or per-column quantization in our setting), the $\ell_\infty$ penalty is imposed column-wise on the weight matrix to reduce the quantization scale of each channel. That is, MagR amounts to solving

$$\min_{\boldsymbol{W} \in \mathbb{R}^{m \times n}} \frac{1}{2}\|\boldsymbol{X}\boldsymbol{W} - \boldsymbol{X}\hat{\boldsymbol{W}}\|_F^2 + \alpha \sum_{j=1}^{n} \|\boldsymbol{w}_j\|_\infty$$

In this case, we take the following iteration:

$$\boldsymbol{W}^{k+1} = \text{prox}_{\eta\alpha\|\cdot\|_\infty}\left(\boldsymbol{W}^k - \eta \cdot \boldsymbol{X}^\top \boldsymbol{X}(\boldsymbol{W}^k - \hat{\boldsymbol{W}})\right),$$

where the proximal operator $\text{prox}_{t\|\cdot\|_\infty}$ and the corresponding projection $\text{proj}_{\|\cdot\|_1 \leq 1}$ in (4) are applied *column-wise* to the matrix input. Hereby we summarize MagR for processing one linear layer in Algorithm 1 with the column-wise $\ell_1$-ball projection as detailed in Algorithm 2, which generalizes Algorithm 3 in Appendix A.1, by handling matrix inputs (or batches of vectors).

---

**Algorithm 1** Per-channel MagR for one linear layer.

---

**Input:** Pre-trained weight matrix $\hat{\boldsymbol{W}} \in \mathbb{R}^{m \times n}$; Hessian matrix $\boldsymbol{H} = \boldsymbol{X}^\top \boldsymbol{X} \in \mathbb{R}^{m \times m}$; max iteration number $K$; step size $\eta = \frac{1}{\lambda_{\max}(\boldsymbol{H})}$; penalty parameter $\alpha > 0$.
**Output:** Preprocessed weights $\boldsymbol{W} \in \mathbb{R}^{m \times n}$.

1: Initialize $\boldsymbol{W}^0 = \hat{\boldsymbol{W}}$.
2: **for** $k = 0, \ldots, K - 1$ **do**
3:     $\boldsymbol{V}^k = \boldsymbol{W}^k - \eta \cdot \boldsymbol{H}(\boldsymbol{W}^k - \hat{\boldsymbol{W}})$                                  gradient descent step
4:     $\boldsymbol{W}^{k+1} = \boldsymbol{V}^k - \eta\alpha \cdot \text{proj}_{\|\cdot\|_1 \leq 1}\left(\frac{\boldsymbol{V}^k}{\eta\alpha}\right)$           $\text{proj}_{\|\cdot\|_1 \leq 1}$ is described in Alg. 2
5: **end for**
6: **return** $\boldsymbol{W} = \boldsymbol{W}^K$

---

**Extension to Per-Group Quantization.** By using more float scaling factors, per-group quantization becomes a preferred strategy for mitigating accuracy loss at extremely low bit-widths. In this approach, a weight vector $\mathbf{w} \in \mathbb{R}^m$ is segmented into groups of weights, each containing $d$ elements, with all weights within a group sharing a common scaling factor for quantization. Here, per-group MagR applies an $\ell_\infty$ penalty to each vector of grouped weights. Consequently, the $\ell_1$-ball projection is independently performed on these vectors, while maintaining the gradient descent step unchanged. We note that the group-wise $\ell_1$-ball projection can be easily done using Algorithm 2, with an additional reshaping of the input $\boldsymbol{V} \in \mathbb{R}^{m \times n}$ into $\mathbb{R}^{d \times \left(\frac{m}{d} \cdot n\right)}$.

**Algorithm 2** Column-wise projection onto the unit $\ell_1$-ball.

---

**Input:** Matrix $\boldsymbol{V} \in \mathbb{R}^{m \times n}$; the radius of $\ell_1$ ball, $\epsilon = 1$.
**Output:** $\boldsymbol{W} \in \mathbb{R}^{m \times n}$ such that all columns $\|\boldsymbol{w}_j\|_1 \leq \epsilon, \forall j \in [n]$.
 1: Create a binary mask $\boldsymbol{M} \in \mathbb{R}^{m \times n}$ filtering out the columns of $\boldsymbol{V}$ with $\|\boldsymbol{v}_j\|_1 \leq \epsilon$.
 2: Sort $|\boldsymbol{V}|$ column-wise in descending order into $\boldsymbol{U}$.
 3: Find index $\rho_j = \max \left\{ i \in [m] : u_{i,j} > \frac{1}{i} \left( \sum_{r=1}^{i} u_{r,j} - \epsilon \right) \right\}, \forall j \in [n]$
 4: Define $\theta_j = \frac{1}{\rho_j} \left( \sum_{r=1}^{\rho_j} u_{r,j} - \epsilon \right), \forall j \in [n]$
 5: Tile $\boldsymbol{\theta} \in \mathbb{R}^n$ into $\boldsymbol{\Theta} \in \mathbb{R}^{m \times n}$ along the row.
 6: Compute $\boldsymbol{W} = (1 - \boldsymbol{M}) \odot \boldsymbol{V} + \boldsymbol{M} \odot \operatorname{sgn}(\boldsymbol{V}) \odot \max\{|\boldsymbol{V}| - \boldsymbol{\Theta}, 0\}$
 7: **return** $\boldsymbol{W}$

---

## 5 Experiments

**Overview.** We tested the proposed MagR for `INT4`, `INT3`, and `INT2` weight quantization. In our notations, the weight and activation bits are denoted by '`W`' and '`A`', respectively. Additionally, we implemented group-wise weight quantization with the group size denoted by '`g`'. For example, W2A16g128 signifies `INT2` weight and `FP16` activation (i.e., `INT2` weight-only quantization) with a group size of 128.

We employed our MagR processing approach on top of the two gradient-free PTQ methods in main text, RTN and OPTQ [16], to quantize the LLaMA1 (7B-65B) [37] and LLaMA2 (7B-70B) [38] model families. In the Appendix A.2, we extend MagR with QuIP [8] (MagR+QuIP) to quantize LLaMA2 (7B-70B) model families. By applying MagR on top of RTN (MagR+RTN), we achieved better results than AWQ [25] for per-channel `INT3` and `INT4` weight quantization. Additionally, MagR combined with OPTQ (MagR+OPTQ) achieved state-of-the-art performance for `INT3` and `INT4` qunatization. To enhance the per-channel `INT2` quantization, we ran 30 additional iterations of coordinate descent algorithm [4, 51] on top of OPTQ, which we denote by MagR+OPTQ$^\dagger$. It turns out MagR+OPTQ$^\dagger$ is superior to both Omniquant [36] and QuIP [8] in terms of perplexity (Table 9), and falls just short of QuIP in zero-shot tasks for 13B and 70B models (Table 4). Note that QuIP uses random orthogonal transformations (so-called Incoherence Processing) to process both the weights and features, resulting in $1.5\times$ slower throughput than OPTQ. In contrast, MagR-based method does not introduce any overhead whatsoever compared with OPTQ.

In conclusion, our MagR-based PTQ method is intuitive yet effective in compressing models into extreme bit-widths, while maintaining performance without introducing any inference overhead.

**Datasets and Evaluation.** Following the previous work [16, 25, 36], we evaluate the quantized model on language generation tasks on WikiText2 [28] and C4 [33]. Additionally, we test its performance on zero-shot tasks, including PIQA [5], ARC (Easy and Challenge) [10], and Winogrande [35]. For the language generation experiments, our implement is based on the OPTQ's [16] repository, which is built using PyTorch. For executing all zero-shot tasks, we adhere to the lm-eval-harness [17].

**Baseline**: For the language generation task, we compare our method with RTN, OPTQ [16], AWQ [25] and OmniQuant [36] on LLaMA1 and LLaMA2 models. In addition to the aforementioned methods, we also conduct a comparison with QuIP [8] on the LLaMA2-70B model. In the zero-shot task, we focus on four individual tasks and compare the average accuracy across all four tasks with Omniquant [36].

**Implementation details.** We utilized the HuggingFace implementations of the LLaMA1 and LLaMA2 models and perform quantization on a single NVIDIA A100 GPU with 80GB of memory. Following the OPTQ method, we load one block consisting of 7 linear layers into GPU memory at a time. In line with previous work [8, 16], the input matrix $\boldsymbol{X}$ is obtained by propagating the calibration data through the quantized layers.

**The choice of parameters.** To ensure that the MagR-processed layer output $\boldsymbol{X}\boldsymbol{W}$ is faithful to the original $\boldsymbol{X}\hat{\boldsymbol{W}}$, we need to use a tiny penalty parameter $\alpha$ in (2). For per-channel quantization, $\alpha$ was fixed to be $10^{-3}$ in our experiments, but we did find that setting it to a smaller value of $5 \times 10^{-4}$ or $10^{-4}$ can sometimes slightly improve the perplexity (with a relative change of $< 1\%$ in ppl). Similarly for per-group quantization, we set $\alpha$ to $10^{-4}$, while reducing it to $5 \times 10^{-5}$ or $10^{-5}$ could

sometimes also slightly improve the perplexity. An ablation study on $\alpha$ is provided in the Appendix A.2.

Furthermore, we used a multiplicative scalar $\beta < 1$ to decay the standard quantization step $\delta = \frac{\max(\boldsymbol{w}) - \min(\boldsymbol{w})}{2^b - 1}$ (or equivalently, the quantization scale) of the quantizer. In other words, our $\delta = \beta \cdot \frac{\max(\boldsymbol{w}) - \min(\boldsymbol{w})}{2^b - 1}$. It has been shown in existing works [21, 34] that, optimal quantization step for binary or ternary quantization yielding the minimum quantization error is not given by $\frac{\max(\boldsymbol{w}) - \min(\boldsymbol{w})}{2^b - 1}$. Shrinking $\delta$ at low bit-width results in a more clustered quantization grid lattice that fits the weights better, which leads to a smaller overall error. In general, $\beta$ is positively correlated with the bit-width used. For per-channel quantization, the best $\beta \in [0.8, 0.85]$ on INT2 quantization, whereas the empirically optimal $\beta$ is around 0.9 for INT3 quantization. As for INT4, $\beta$ is simply set to 1, that is, we used the standard quantization step. In addition, for per-group quantization, we chose $\beta = 0.95$ for both INT2 and INT3 quantization. The ablation study of $\beta$ is in the Appendix A.2. We observed that this refinement on the quantization step $\delta$ significantly improves the performance of the PTQ method. In addition, the iteration number $K$ in Algorithm 1 was set to 150 across all the experiments.

Table 3: **Perplexity of quantized LLaMA2 models on Wikitext2 and C4**. We report WikiText2 and C4 perplexity in this table. LLaMA1 resutls can be found in the Appendix.

| Datasets | | Wikitext2 | | | C4 | | |
|---|---|---|---|---|---|---|---|
| **LLaMA / PPL↓** | | 2-7B | 2-13B | 2-70B | 2-7B | 2-13B | 2-70B |
| FP16 | Baseline | 5.47 | 4.88 | 3.31 | 6.97 | 6.46 | 5.52 |
| W2A16 | OPTQ | 7.7e3 | 2.1e3 | 77.95 | NAN | 323.12 | 48.82 |
| | OmniQuant | 37.37 | 17.21 | 7.81 | 90.64 | 26.76 | 12.28 |
| | QuIP | 27.13 | **10.09** | 6.33 | 31.33 | **13.13** | 8.94 |
| | MagR+OPTQ[†] | **16.73** | 11.14 | **5.95** | **23.73** | 14.45 | **8.53** |
| W2A16 g128 | OPTQ | 36.77 | 28.14 | - | 33.70 | 20.97 | - |
| | OmniQuant | 11.06 | 8.26 | 6.55 | 15.02 | 11.05 | 8.52 |
| | MagR+OPTQ | **9.94** | **7.63** | **5.52** | **14.08** | **10.57** | **8.05** |
| W3A16 | RTN | 539.48 | 10.68 | 7.52 | 402.35 | 12.51 | 10.02 |
| | OPTQ | 8.37 | 6.44 | 4.82 | 9.81 | 8.02 | 6.57 |
| | AWQ | 24.00 | 10.45 | - | 23.85 | 13.07 | - |
| | OmniQuant | 6.58 | 5.58 | 3.92 | 8.65 | 7.44 | 6.06 |
| | QuIP | 6.50 | **5.34** | 3.85 | 8.74 | 7.34 | 6.14 |
| | MagR+RTN | 8.66 | 6.55 | 4.64 | 10.78 | 8.26 | 6.77 |
| | MagR+OPTQ | **6.41** | 5.41 | **3.82** | **8.23** | **7.19** | **6.03** |
| W3A16 g128 | RTN | 6.66 | 5.51 | 3.97 | 8.40 | 7.18 | 6.02 |
| | OPTQ | 6.29 | 5.42 | 3.85 | 7.89 | 7.00 | 5.85 |
| | AWQ | 6.24 | 5.32 | - | 7.84 | 6.94 | - |
| | OmniQuant | 6.03 | 5.28 | 3.78 | **7.75** | 6.98 | 5.85 |
| | MagR+RTN | 6.46 | 5.45 | 3.95 | 8.22 | 7.12 | 6.00 |
| | MagR+OPTQ | **6.00** | **5.23** | **3.71** | 7.77 | **6.93** | **5.84** |
| W4A16 | RTN | 6.11 | 5.20 | 3.67 | 7.71 | 6.83 | 5.79 |
| | OPTQ | 5.83 | 5.13 | 3.58 | 7.37 | 6.70 | 5.67 |
| | AWQ | 6.15 | 5.12 | - | 7.68 | 6.74 | - |
| | OmniQuant | 5.74 | 5.02 | 3.47 | 7.35 | 6.65 | 5.65 |
| | QuIP | 5.94 | 5.01 | 3.53 | 8.01 | 6.88 | 5.87 |
| | MagR+RTN | 5.91 | 5.17 | 3.58 | 7.52 | 6.81 | 5.72 |
| | MagR+OPTQ | **5.70** | **4.97** | **3.44** | **7.28** | **6.63** | **5.63** |

## 5.1 Language Generation

We concentrate our analysis on perplexity-based tasks. The results for the LLaMA2 family with context length of 2048, are elaborated in Table 9, while those for LLaMA1 are provided in Appendix Table 6. As evidenced by the tables, the MagR preprocessing consistently improve the performance of the baselines RTN and OPTQ. Moreover, MagR+OPTQ consistently outperforms most baseline across the LLaMA family models for both per-channel and per-group weight quantization. Particularly, for

`INT2`, MagR+OPTQ[†] performs 30 additional coordinate descent (CD) iterations [4, 51] on top of OPTQ to refine the solution, surpassing all baselines.

Furthermore, MagR+RTN achieves performance comparable to OPTQ. Notably, it outperforms AWQ by a significant margin in `INT3` quantization, implying that MagR proves more effective as a preprocessing method compared to channel-wise scaling.

Table 4: **Multi-task results of quantized LLaMA2 models.** This table reports the accuracy of 4 zero-shot tasks. Perplexity results can be found in the Appendix.

| LLaMA2 / Acc↑ | WBits | Method | ARC-C | ARC-E | PIQA | Winogrande | **Avg.** |
|---|---|---|---|---|---|---|---|
| | FP16 | - | 40.0 | 69.3 | 78.5 | 67.3 | 63.8 |
| | 4 | OmniQuant | 37.9 | 67.8 | 77.1 | **67.0** | 62.5 |
| | 4 | MagR+OPTQ | **39.3** | **68.4** | **78** | 66.5 | **63.1** |
| LLaMA2-7B | 3 | OmniQuant | **35.3** | **62.6** | 73.6 | **63.6** | **58.8** |
| | 3 | MagR+OPTQ | 34.6 | 62 | **74.7** | 63 | 58.6 |
| | 2 | OmniQuant | 21.6 | 35.2 | 57.5 | **51.5** | 41.5 |
| | 2 | QuIP | 19.4 | 26.0 | 54.6 | 51.8 | 37.5 |
| | 2 | MagR+OPTQ[†] | **22.0** | **36.7** | **59.8** | 51.1 | **42.4** |
| | FP16 | - | 45.6 | 73.3 | 79.1 | 69.6 | 66.9 |
| | 4 | OmniQuant | 43.1 | 70.2 | 78.4 | 67.8 | 64.9 |
| | 4 | QuIP | **44.9** | **73.3** | **79** | **69.7** | **66.7** |
| | 4 | MagR+OPTQ | 44.2 | 72.0 | 78.0 | 68.6 | 65.7 |
| LLaMA2-13B | 3 | OmniQuant | 42.0 | 69.0 | 77.7 | 65.9 | 63.7 |
| | 3 | QuIP | 41.5 | **70.4** | 76.9 | **69.9** | **64.7** |
| | 3 | MagR+OPTQ | **42.2** | 69.0 | **77.7** | 66.5 | 63.9 |
| | 2 | OmniQuant | 23.0 | 44.4 | **62.6** | 52.6 | 45.7 |
| | 2 | QuIP | **23.5** | **45.2** | 62.0 | **52.8** | **45.9** |
| | 2 | MagR+OPTQ[†] | 23.2 | 44.3 | 62.4 | 52.1 | 45.5 |
| | FP16 | - | 51.1 | 77.7 | 81.1 | 77.0 | 71.7 |
| | 4 | OmniQuant | 49.8 | **77.9** | 80.7 | 75.8 | 71.1 |
| | 4 | QuIP | 47.0 | 74.3 | 80.3 | 76.0 | 69.4 |
| | 4 | MagR+OPTQ | **50.1** | 77.5 | **80.8** | **76.0** | **71.1** |
| LLaMA2-70B | 3 | OmniQuant | 47.6 | 75.7 | 79.7 | 73.5 | 69.1 |
| | 3 | QuIP | 46.3 | 73.2 | **80.0** | 74.6 | 68.5 |
| | 3 | MagR+OPTQ | **47.7** | **76.6** | 79.4 | **75.4** | **69.8** |
| | 2 | OmniQuant | 28.7 | 55.4 | 68.8 | 53.2 | 51.5 |
| | 2 | QuIP | 34.0 | **62.2** | **74.8** | **67.5** | **59.6** |
| | 2 | MagR+OPTQ[†] | **35.9** | 61.3 | 74.7 | 64.8 | 59.2 |

## 5.2 Zero-Shot Tasks

We evaluated the performance of quantized models on several zero-shot tasks. The results are reported in Table 4. Similar to previous observations, the proposed MagR demonstrates superior performance on most models compared to OmniQuant, with a small gap compared to QuIP [8]. Nonetheless, it is reasonable and commendable that our algorithm achieves results close to QuIP without introducing any inference overhead. It is possible to further improve our approach based on the insight behind QuIP [8] — i.e., quantization benefits from incoherent weight and Hessian matrices; see Table 9 for the results in the appendix.

## 5.3 Preprocessing and Quantization Runtime

We report the execution time of MagR+RTN and MagR+OPTQ on a single NVIDIA A100 GPU in Table 5. For example, it typically took 0.5-7.5 hours for MagR+OPTQ to quantize the LlaMA2 models. We note that the integration of MagR can markedly enhance the performance of the standard OPTQ [16]. It is noted that MagR+OPTQ[†] for `INT2` weight quantization requires a longer runtime

Table 5: **The runtime of MagR+RTN, MagR+OPTQ, and MagR+OPTQ$^\dagger$ on an Nvidia A100 GPU, with comparisons to their vanilla counterparts, namely, RTN and OPTQ.**

| Method/ Model | LLaMA2-7B | LLaMA2-13B | LLaMA2-70B |
|---|---|---|---|
| RTN | 5 min | 12 min | 36 min |
| MagR+RTN | 20 min | 40 min | 4 hr |
| OPTQ | 22 min | 40 min | 4 hr |
| MagR+OPTQ | 35 min | 70 min | 7.5 hr |
| MagR+OPTQ$^\dagger$ | 2.5 hr | 5.5 hr | 31 hr |

due to the additional CD iterations, extending the quantization process for LLaMA2-70B to 31 hr. It also reveals that the preprocessing overhead for quantizing the LLaMA2 models (7B-70B) amounts to approximately 15 min, 30 min, and 3.5 hr, respectively. In comparison, our total runtime is roughly half of that of the gradient-based method, OmniQuant [36], while achieving at least comparable results. Moreover, MagR introduces no post-processing step or overhead during inference.

## 6 Concluding Remarks

In this paper, we proposed MagR, based on $\ell_\infty$-regularization, to significantly reduce the maximum weight magnitude of pre-trained LLMs within each layer while preserving their output. MagR is designed to enhance the accuracy of backpropagation-free PTQ methods that use layer-wise reconstruction, such as RTN and OPTQ. MagR produces a more clustered distribution of weights and leads to a smaller quantization step, thereby facilitating the subsequent PTQ task. To solve the $\ell_\infty$-regularization problem, we used the classical proximal gradient descent algorithm with $\ell_1$-ball projections, tailored to handle matrix variables efficiently. Our experiments on LLaMA family validated the effectiveness of the MagR approach, achieving the state-of-the-art performance on NLP tasks. Remarkably, unlike existing weight preprocessing techniques that require performing an inverse transformation on features during inference, MagR eliminates the need for post-processing and incurs no overhead. This renders MagR more practical for the deployment of quantized models.

## Acknowledgement

This work was partially supported by NSF grants DMS-2208126, DMS-2110836, IIS-2110546, CCSS-2348046, SUNY-IBM AI Research Alliance Grant, and a start-up grant from SUNY Albany. We would also like to thank SUNY Albany for providing access to the Nvidia A100 GPUs.

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

# A    Appendix / supplemental material

## A.1    Projection of Vectors Onto $\ell_1$-Ball

In this section, we show how to compute the projection onto the unit $\ell_1$-Ball. That is, for any fixed $\boldsymbol{v} \in \mathbb{R}^m$, we solve the optimization problem:

$$\min_{\boldsymbol{x} \in \mathbb{R}^m} \|\boldsymbol{x} - \boldsymbol{v}\|^2 \quad \text{subject to} \quad \|\boldsymbol{x}\|_1 \leq 1. \tag{5}$$

Consider the Lagrangian $\mathcal{L}(\boldsymbol{x}, \lambda) = \frac{1}{2}\|\boldsymbol{x} - \boldsymbol{v}\|^2 + \lambda(\|\boldsymbol{x}\|_1 - 1)$, where $\lambda \in \mathbb{R}$ is the lagrange multiplier. Let $\boldsymbol{x}^*$ be the optimal solution of (5), then there exits $\lambda^*$ such that the following Karush–Kuhn–Tucker (KKT) conditions [6] hold:

- Stationarity: $\boldsymbol{0} \in \partial_{\boldsymbol{x}} \mathcal{L}(\boldsymbol{x}^*, \lambda^*) \Leftrightarrow \boldsymbol{x}^* = \operatorname{sgn}(\boldsymbol{v}) \odot \max\{|\boldsymbol{v}| - \lambda^*, 0\}$
- Slackness: $\lambda^* (\|\boldsymbol{x}^*\|_1 - 1) = 0 \Leftrightarrow \lambda^* = 0$ or $\|\boldsymbol{x}^*\|_1 = 1$.
- Primal feasibility: $\|\boldsymbol{x}^*\|_1 - 1 \leq 0 \Leftrightarrow \|\boldsymbol{x}^*\|_1 \leq 1$
- Dual feasibility: $\lambda^* \geq 0 \Leftrightarrow \lambda^* = 0$ or $\lambda^* > 0$

where sgn is the signum function applied element-wise on vectors, i.e.,

$$\operatorname{sgn}(\boldsymbol{v})_i = \begin{cases} 1 & \text{if } v_i > 0, \\ 0 & \text{if } v_i = 0, \\ -1 & \text{if } v_i < 0. \end{cases}$$

We note that the stationarity condition:

$$\boldsymbol{x}^* = \operatorname{sgn}(\boldsymbol{v}) \odot \max\{|\boldsymbol{v}| - \lambda^*, 0\}$$

gives the projection of $\boldsymbol{v}$ onto the unit $\ell_1$ ball, provided $\lambda^*$ is known. Therefore, what remains is to find $\lambda^*$:

- Case I: $\lambda^* = 0$. We have $\boldsymbol{x}^* = \boldsymbol{v}$, which corresponds to the case that $\|\boldsymbol{v}\|_1 \leq 1$.
- Case II: $\lambda^* > 0$. The slackness condition yields $\|\boldsymbol{x}^*\|_1 = 1$, i.e., $\sum_{i=1}^n |x_i^*| = 1$, or equivalently, $\sum_{i=1}^n |\operatorname{sgn}(v_i)(|v_i| - \lambda^*)_+| = \sum_{i=1}^n (|v_i| - \lambda^*)_+ = 1$ with $x_+ := \max\{x, 0\}$. That is, $\lambda^*$ is the root of the piece-wise linear equation:

$$\sum_{i=1}^n (|v_i| - \lambda)_+ = 1, \tag{6}$$

which can be solved by sorting.

In summary, Algorithm 3 details the implementation of projecting $\boldsymbol{v} \in \mathbb{R}^m$ onto a general $\ell_1$-ball with radius $\epsilon$. In Step 4, we specifically compute the root $\lambda^*$ (or $\theta$) of (6) for Case II.

---

**Algorithm 3** Projection onto $\ell_1$-ball.

---

**Input:** Vector $\boldsymbol{v} \in \mathbb{R}^m$; the radius of $\ell_1$ ball, $\epsilon = 1$.
**Output:** $\boldsymbol{w} \in \mathbb{R}^m$ such that $\|\boldsymbol{w}\|_1 \leq \epsilon$.

1: **if** $\|\boldsymbol{v}\|_1 > \epsilon$ **then**
2:     Sort $|\boldsymbol{v}|$ into $\boldsymbol{\mu}$ such that $\mu_1 \geq \mu_2 \geq \ldots \geq \mu_m$.
3:     Find index $\rho = \max\left\{i \in [m] : \mu_i > \frac{1}{i}\left(\sum_{r=1}^i \mu_r - \epsilon\right)\right\}$
4:     Define $\theta = \frac{1}{\rho}\left(\sum_{r=1}^\rho \mu_r - \epsilon\right)$
5:     Compute $\boldsymbol{w} = \operatorname{sgn}(\boldsymbol{v}) \odot \max\{|\boldsymbol{v}| - \theta, 0\}$
6: **else**
7:     $\boldsymbol{w} = \boldsymbol{v}$
8: **end if**
9: **return** $\boldsymbol{w}$

---

## A.2    Additional Experimental Results

Table 6 shows the results for WikiText2 and C4 perplexity on the LLaMA1.

Table 6: **Weight-only quantization Results of WikiText2 and C4 on LLaMA1 Models.**

| Datasets | | Wikitext2 | | | | C4 | | | |
|---|---|---|---|---|---|---|---|---|---|
| **LLaMA / PPL↓** | | 1-7B | 1-13B | 1-30B | 1-65B | 1-7B | 1-13B | 1-30B | 1-65B |
| FP16 | | 5.68 | 5.09 | 4.10 | 3.53 | 7.08 | 6.61 | 5.98 | 5.62 |
| W2A16 | OPTQ | 2.1e3 | 5.5e3 | 499.75 | 55.91 | 689.13 | 2.5e3 | 169.80 | 40.58 |
| | OmniQuant | **15.47** | 13.21 | 8.71 | 7.58 | 24.89 | 18.31 | 13.89 | 10.77 |
| | MagR+OPTQ$^\dagger$ | 19.98 | **9.41** | **8.47** | **6.41** | **24.69** | **16.37** | **13.09** | **8.82** |
| W2A16 g128 | OPTQ | 44.01 | 15.60 | 10.92 | 9.51 | 27.71 | 15.29 | 11.93 | 11.99 |
| | OmniQuant | **9.72** | **7.93** | 7.12 | **5.95** | **12.97** | **10.36** | 9.36 | **8.00** |
| | MagR+OPTQ | 9.89 | 9.22 | **6.72** | 6.41 | 13.14 | 10.62 | **8.05** | 9.14 |
| W3A16 | RTN | 25.73 | 11.39 | 14.95 | 10.68 | 28.26 | 13.22 | 28.66 | 12.79 |
| | OPTQ | 8.06 | 6.76 | 5.84 | 5.06 | 9.49 | 8.16 | 7.29 | 6.71 |
| | AWQ | 11.88 | 7.45 | 10.07 | 5.21 | 13.26 | 9.13 | 12.67 | 7.11 |
| | OmniQuant | **6.49** | 5.68 | 4.74 | **4.04** | **8.19** | 7.32 | 6.57 | **6.07** |
| | MagR+RTN | 7.93 | 6.71 | 5.66 | 4.79 | 9.77 | 8.46 | 7.38 | 6.87 |
| | MagR+OPTQ | 6.86 | **5.43** | **4.73** | 4.2 | 8.65 | **7.21** | **6.56** | 6.16 |
| W3A16 g128 | RTN | 7.01 | 5.88 | 4.87 | 4.24 | 8.62 | 7.49 | 6.58 | 6.10 |
| | OPTQ | 6.55 | 5.62 | 4.80 | 4.17 | 7.85 | 7.10 | 6.47 | 6.00 |
| | AWQ | 6.46 | 5.51 | 4.63 | 3.99 | 7.92 | 7.07 | 6.37 | 5.94 |
| | OmniQuant | **6.15** | 5.44 | 4.56 | **3.94** | **7.75** | **7.05** | 6.37 | 5.93 |
| | MagR+RTN | 6.90 | 5.50 | 4.82 | 4.17 | 8.46 | 7.19 | 6.52 | 6.02 |
| | MagR+OPTQ | 6.29 | **5.41** | **4.52** | 3.95 | 7.78 | 7.09 | 6.38 | **5.93** |
| W4A16 | RTN | 6.43 | 5.55 | 4.57 | 3.87 | 7.93 | 6.98 | 6.34 | 5.85 |
| | OPTQ | 6.13 | 5.40 | 4.48 | 3.83 | 7.43 | 6.84 | 6.20 | 5.80 |
| | AWQ | 6.08 | 5.34 | 4.39 | 3.76 | 7.52 | 6.86 | 6.17 | 5.77 |
| | OmniQuant | **5.86** | **5.21** | 4.25 | **3.71** | **7.34** | **6.76** | **6.11** | **5.73** |
| | MagR+RTN | 6.16 | 5.42 | 4.36 | 3.80 | 7.66 | 6.87 | 6.22 | 5.82 |
| | MagR+OPTQ | 6.03 | 5.23 | **4.24** | 3.72 | 7.39 | 6.77 | 6.13 | 5.75 |

Table 7: The perplexity of quantized LLaMa2-7B models for different $\alpha$ values.

| $\alpha$ | W/A | Wikitext2 (PPL) | C4 (PPL) |
|---|---|---|---|
| 0.005 | 4/16 | 5.84 | 7.55 |
| **0.001** | 4/16 | **5.70** | **7.28** |
| 0.0005 | 4/16 | 5.72 | 7.29 |
| 0.0001 | 4/16 | 5.78 | 7.35 |
| 0.00001 | 4/16 | 5.81 | 7.40 |
| 0.005 | 3/16 | 6.64 | 8.74 |
| **0.001** | 3/16 | **6.41** | **8.23** |
| 0.0005 | 3/16 | 6.49 | 8.38 |
| 0.0001 | 3/16 | 6.83 | 8.79 |
| 0.00001 | 3/16 | 7.08 | 9.19 |

## A.3 Ablation Study

**Impact of the parameter** $\alpha$**.** The tiny penalty parameter $\alpha$ balances the trade-off between output discrepancy and the maximum magnitude of the weights. We carry out experiments on channel-wise quantization for differernt $\alpha$ on LLaMA2-7B. The choice of $\alpha$ is independent of the bit-width. As shown in Table 7, we can find that both too large and too small $\alpha$ will lead performance degeneration. Compared to INT4, fluctuations in alpha at INT3 result in greater performance fluctuations. Fortunately, $\alpha = 0.001$ works well for all channel-wise quantization.

**Impact of the parameter** $\beta$**.** We shrink the quantization step to reduce the overall quantization error by a multiplicative scalar $\beta$. To investigate the influence of $\beta$, we experiment with different value of $\beta$ at INT3 and INT2 channel-wise quantization. As shown in Table 8, $\beta$ is positively correlated with the bit-width. Specifically, the best $\beta$ is around 0.9 for INT3 quantization and for INT2 quantization the optimal $\beta$ is around 0.8.

Table 8: The perplexity of quantized LLaMa2-7B models for different $\beta$ values.

| $\beta$ | W/A | Wikitext2 (PPL) | C4 (PPL) |
|---|---|---|---|
| 1 | 3/16 | 6.43 | 8.33 |
| **0.9** | 3/16 | **6.41** | **8.23** |
| 0.85 | 3/16 | 6.48 | 8.39 |
| 0.8 | 3/16 | 7.08 | 9.19 |
| 1 | 2/16 | 16.99 | 24.12 |
| 0.9 | 2/16 | 20.88 | 31.78 |
| 0.85 | 2/16 | 16.76 | 24.45 |
| **0.8** | 2/16 | **16.73** | **23.73** |

**Impact of MagR on Quantization Error.** To explore how MagR affects quantization error, we compared the errors with and without MagR by randomly select five layers from LLaMA2 models. As illustrated in Figure 2, quantization error is notably reduced across all layers with the application of MagR.

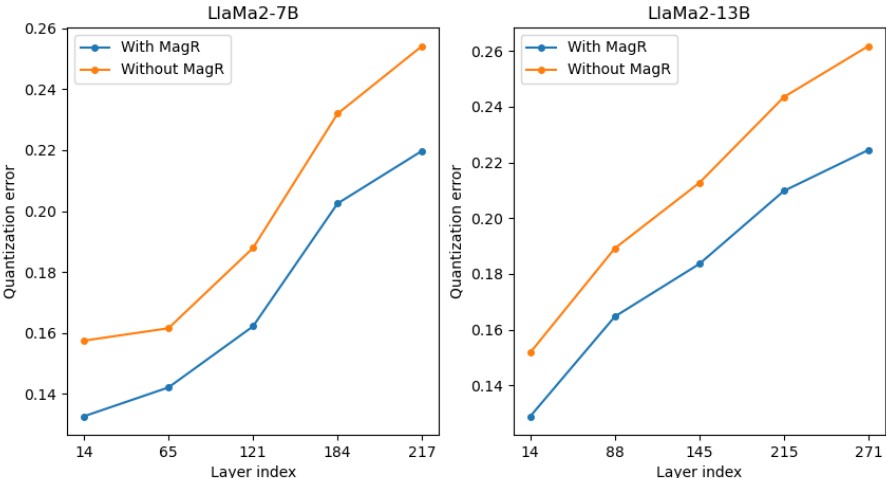

Figure 2: Layer-wise quantization errors (root mse) for MagR+OPTQ and OPTQ, respectively, for 4-bit quantization. The layers are selected randomly for visualization, but improvement is consistent across all layers.

**The adaptive capacity of MagR.** We investigated the combined effects of MagR and QuIP. As illustrated in Table 9, incorporating MagR significantly enhances the performance of QuIP, leading to improved quantization results for the LLaMA2 models family.

Table 9: **Perplexity of MagR+QuIP for LLaMA2 models on Wikitext2 and C4**.

| Datasets | | Wikitext2 | | C4 | |
|---|---|---|---|---|---|
| **LLaMA / PPL↓** | | 2-7B | 2-13B | 2-7B | 2-13B |
| FP16 | Baseline | 5.47 | 4.88 | 6.97 | 6.46 |
| W2A16 | QuIP | 27.13 | 10.09 | 31.33 | 13.13 |
| | MagR+QuIP | 13.31 | 9.40 | 14.49 | 11.07 |
| W3A16 | QuIP | 6.50 | 5.34 | 8.74 | 7.34 |
| | MagR+QuIP | 6.25 | 5.29 | 7.88 | 7.02 |
| W4A16 | QuIP | 5.94 | 5.01 | 8.01 | 6.88 |
| | MagR+QuIP | 5.74 | 4.99 | 7.25 | 6.63 |

