# OpenReview forum: "MagR: Weight Magnitude Reduction for Enhancing Post-Training Quantization"
_NeurIPS.cc/2024/Conference — NeurIPS 2024 poster_

### Official Review · Reviewer_5GSj · 2024-07-10

**Soundness:** 3
**Presentation:** 2
**Contribution:** 3
**Rating:** 8
**Confidence:** 4

**Summary:**

- The authors propose an optimization-based preprocessing technique called Weight Magnitude Reduction (MagR) to improve the performance of post-training quantization (PTQ) for large language models (LLMs). They motivate their method from previous work, showing that linear transformation of weights can render more quantization-friendly, allowing for sub-4bit quantization. However, such methods normally require a post-processing step during inference to undo that linear transformation, introducing a run-time overhead.
- Instead, they propose a non-linear transformation without needing an additional post-processing step. That transformation aims to minimise the $\ell_{\infty}$ norm, or the maximum value of the weights per channel or column while preserving the layer's output. By making the weight distribution more compact around zero and removing outliers, they are able to quantize weights to 2 or 3 bits with little accuracy drop compared to competing methods.
- They formulate the optimization as an $\ell_{\infty}$ -regularisation and solve it using proximal gradient descent. In fact, the problem reduces to computing the projection of weight vectors onto the $\ell_1$ ball, for which there are several efficient algorithms available.

**Strengths:**

- MagR achieves state-of-art weight quantization without introducing any additional computation overhead. This is a very important contribution as all other competing methods with the linear transformation of weight, such as QuIP, require a linear transformation of the transformer block input/output. Having said this, the inference runtime comparison to QuIP is missing from the experiments section
- The authors extensively evaluate the methods, evaluating a large corpus of datasets and benchmarks. They also prove that they outperform strong competing methods, such as OmniQuant, in almost all cases.
- The idea of regularizing the weight's $\ell_{\infty}$ norm is quite novel and very effective. In fact, I am surprised this has not been established before in PTQ literature.

**Weaknesses:**

- The paper is very focused on ppl and accuracy results on a wide range of benchmarks and datasets at the cost of other important ablation studies. For example:
	- ablation study on the penalty factor $\alpha$ and its impact on the final accuracy.
	- Inference runtime comparison is missing compared to competing methods, such as QuIP. In fact, the main claim of the paper is that it is faster than QuIP, but there are no studies supporting it.
	- Results with activation quantization beyond FP16 and runtime improvements vs baselines.
- The connection between the rank-deficient feature matrix argument and $\ell_{\infty}$ regularisation requires further theoretical justification. There must be a theoretical guarantee for the optimality of that optimizaiton. In fact, the method would be stronger if it was formulated more generally as an $\ell_p$ constrained, and the authors show that $\ell_{\infty}$ is better constrained.

**Questions:**

- Table 1: It is unclear to me how all these numbers are relevant in supporting the method. One can see from the maximum value that all Llama Models are rank-deficient based on the maximum value being below 100%. How are the rest of the statistics informative?
- Line 124-125: From all the solutions available, why choose the lowest infinity norm? Why is this optimal? What is missing there is the optimality of this norm for quantization purposes. There, in fact, a discussion about quantization noise as a perturbation bounded by $\ell_{\infty}$ -norm in a paper by M. Alizadeh et. al called "Gradient $\ell_1$ regularization for quantization robustness"
- Table 3: activation quantization bitwidth missing
- Please provide ablation studies about the penalty factor $\alpha$. What is the effect of it not being tiny? How sensitive are the results?
- To fully support the claims of the paper, it would be nice to see an ablation study on other norms as regularization. Otherwise, a theoretical justification is required about the optimality of the $\ell_{\infty}$ -norm.
- Paragraph starting at line 210: it is unclear how you establish the optimality of the shrinking factor $\beta$. In fact, I cannot see why you would not search for the optimal clipping threshold based on minimizing the MSE per column or group. This is a standard method for weight clipping in LLM quantization literature.
- Section 5.3: the section's title is misleading. Maybe rename it to preprocessing runtime? Most commonly, runtime refers to the inference runtime
- A section on inference runtime comparison to QuIP is required. The main claim of the paper is that the method is faster than QuIP due to the lack of transformation, but no experimental evidence is provided to support it.

**Limitations:**

The paper lacks a thorough discussion of each limitation. The only reference I found was in lines 245-6 about extending MagR into incoherence processing.

---

> ### Author Rebuttal · Authors · 2024-08-03
>
> Thank you for your constructive comments! We'll discuss the reference by Alizadeh et. al in the revised paper.
>
> The primary concerns were regarding:
>
> **Weaknesses:**
>
>
>   - **The use of** $\ell_\infty$-**norm**: MagR aims to reduce the range of the pre-trained weights (not the quantization noise), and $\ell_\infty$-norm is naturally the best to serve this purpose. So the nature of MagR is different than $\ell_1$-gradient regularization proposed by Alizadeh et. al. Given that the quantization step (or float scalar) is linearly proportional to the range of the pre-trained weights for a fixed bit-width (by the definition of uniform quantizer), **MagR results in a smaller quantization step. This, in turn, leads to a smaller quantization error**. In fact, we can show that the $\ell_2$ quantization error is $O(\delta)$.
> Our additional experiments also demonstrate that **the layer-wise quantization errors indeed are reduced by applying MagR**; see the figure in the attached pdf file. Another notable advantage of $\ell_\infty$-norm over the general $\ell_p$-norm is its closed form of proximal operator, which is the core of the algorithm design for MagR.
>
>   - **Activation quantization**: By its nature, MagR is designed to improve the performance of weight quantization without introducing inference overhead. So we evaluated our method for only weight quantization. We intend to investigate activation quantization in our future work.
>
> - **Ablation study on $\alpha$**:  $\alpha$ is the parameter balancing the tradeoff between the output discrepancy and the max magnitude of the weights. We show the ablation study on $\alpha$ for channel-wise quantization on LLaMA2-7B as below. Note that **the choice of $\alpha$ does not depend on the bit-width** and $\alpha=0.001$ is the best choice for channel-wise quantization.
>
>   | Model   | $\alpha$ | W/A | Wiki (PPL)       | C4 (PPL) |
>   |----------|-|---------|-------|--------------|
>   | LlaMa2-7B    | 0.005 | 4/16 | 5.84     | 7.55     |
>   |                       | **0.001** | 4/16 | **5.70**   |  **7.28**    |
>   |                       | 0.0005 | 4/16 | 5.72     | 7.29     |
>   |                       | 0.0001 | 4/16 | 5.78     |  7.35     |
>   |                       | 0.00001 | 4/16 | 5.81     | 7.40     |
>   |     |     |       |          |
>   |                       | 0.005 | 3/16 | 6.64     | 8.74     |
>   |                       | **0.001** | 3/16 | **6.41**     |  **8.23**    |
>   |                       | 0.0005 | 3/16 | 6.49    | 8.38     |
>   |                       | 0.0001 | 3/16 | 6.83    |  8.79     |
>   |                       | 0.00001 | 3/16 | 7.08  | 9.19      |
>
>
>
> **Questions:**
>
>   - **Fraction ranks in Table 1**: The statistics in Table 1 not only show that all feature matrices are nearly rank-deficient, but also gives us an idea of how low-rank they are and the overall distribution of their fraction ranks across the architectures. Intuitively, MagR works better for low-rank feature matrix because its kernel space would be larger (of a higher dimension), and MagR potentially produces weights with smaller $\ell_\infty$-norm.
>
> - **The choice of $\beta$**:  Thanks for your suggestion. Since our main focus is on MagR preprocessing to reduce the quantization error, we prioritized simplicity and efficiency of the quantization algorithm by fixing the $\beta$, which has proven to be effective.  We agree that optimizing the quantization error with respect to $\beta$ could potentially further improve the performance, and we intend to investigate this in the revision.
>
> - **Preprocessing runtime**: Thanks for your suggestion, we will rename it.
>
> - **Inference time**: MagR essentially replaces the original pre-trained weights with a new set of weights that have a smaller magnitudes prior to actual quantization, without sacrificing the original accuracy or altering the architecture. Consequently, our MagR+OPTQ achieves **exactly the same inference efficiency** as OPTQ. This is immediately supported by the widely-used inference kernel from the AutoGPTQ library. In contrast, at inference time, QuIP requires performing a random linear transformation on the activations before multiplying the quantized weights. Since the code for QuIP's inference is not released, we cannot compare them directly.  But according to QuIP's own report, QuIP's inference speed is at least **1.5 times slower than OPTQ** for the OPT-66B model (81 ms vs 53 ms).

---

> > ### Comment · Reviewer_5GSj · 2024-08-12
> > **Reviewed score**
> >
> > Having reviewed the responses, ablation studies and latest results in the conversation with reviewer jujD, I have decided to increase my score to strong accept, provided the additional results and analyses are included in the camera-ready version.
> >
> > From my initial review, I found the method itself innovative but concerns about the presentation and certain claims. The authors have been diligent in addressing the points raised by me and other reviewers.

---

> > > ### Author Response · Authors · 2024-08-12
> > > **Thanks for the reply**
> > >
> > > We sincerely appreciate your valuable feedback and your recognition of the novelty of our work!
> > >
> > > Thanks again!

---

### Official Review · Reviewer_HDx9 · 2024-07-11

**Soundness:** 3
**Presentation:** 3
**Contribution:** 3
**Rating:** 6
**Confidence:** 4

**Summary:**

The paper proposes a novel approach called Weight Magnitude Reduction (MagR) to improve the performance of PTQ for LLM. The MagR reduces the magnitude of the weights in PTQ. The experiments demonstrate the effectiveness of MagR.

**Strengths:**

1.	The idea is clear.
2.	The paper is easy to read (except typos/errors).

**Weaknesses:**

1.	There exist some typos/grammatical errors in the paper and should be revised.
2.	The presentation is not clear. For example, this paper aims to reduce the parameters of LLM using PTQ, however, the title and abstract did not reflect the contribution of LLM compression. In the abstract, the author claims MagR can diminish the maximum magnitude of the weights and smooth out outliers, however, the title only focuses on Weight Magnitude Reduction.
3.	Provide a more thorough discussion of the generalization ability, robustness, and potential applications of the proposed approach.
4.	The format of references is not correct, i.e., [1], [3], [9],[14], etc.
5.	The main limitation of this paper is that the proposed method lacks theoretical analysis.
6.	While the abstract provides an overview of the proposed method, some aspects of the methodology could benefit from further elaboration in the main paper. Providing step-by-step explanations and intuitive visualizations for key components such as L infinity norm, and L infinity-> L 1 norm, would enhance the reader's understanding.
7.	The paper claims that the method outperforms state-of-the-art methods, but a more comprehensive comparative analysis is needed. Detailed comparisons with other existing approaches, along with discussions about the reasons behind the performance differences, would strengthen the argument for the superiority.
8.	Exploring the reasons behind the success of these techniques and providing intuitive explanations would contribute to the overall scientific contribution of the work.

**Questions:**

1. The paper contains numerous hyperparameters, such as \(\alpha\). Including more ablation studies would enhance the readers' understanding.
2. In Table 3, why did MagR+OPTQ not achieve better results compared to peer competitors?
3. How can the impact of outliers on PQT for LLM be reduced?

---

> ### Author Rebuttal · Authors · 2024-08-02
>
> Thank you for the review! We'll add ablation study and fix the typos and incorrect format as suggested, and include the details for deriving Alg. 1.
>
> Other concerns were regarding:
>
> **Weaknesses:**
> - **Theoretical analysis**: Our new results establish that:
>   - The layer-wise $\ell_2$ quantization error is $O(\delta)$, where $\delta$ is the quantization step of the uniform quantizer. This motivates our preprocessing method based on $\ell_\infty$-norm of the weights since minimizing $\ell_\infty$-norm amounts to minimizing $\delta$.
>   - We can also prove the convergence rate of the proposed MagR algorithm for the $\ell_\infty$-regularization. Specifically, $f(w^k) - f(w^*) = O(1/k) \to 0$ as $k\to \infty$, where $f(w) = \frac{1}{2}|| Xw - X\hat{w}||^2 + \alpha ||w||_{\infty}$ is the objective function, $w^k$ is the $k$-th iterate generated by MagR, and $w^*$ is the true minimizer.
>   - The layer-wise output $\ell_2$-error after MagR preprocessing obeys $|| Xw^* - X\hat{w}|| = O(\sqrt{\alpha})$, where $\alpha>0$ is the penalty parameter.
>
> - **Why MagR works:**
>   - MagR effectively reduces the range of the pre-trained weights by employing $\ell_\infty$-minimization, as illustrated by Figure 1. Given that the quantization step (or float scalar) is linearly proportional to the range of the pre-trained weights for a fixed bit-width (by the definition of uniform quantizer), **MagR results in a smaller quantization step. This, in turn, leads to a smaller quantization error**. In fact, given the activations $X\in\mathbb{R}^{m\times n}$, quantizer $\mathcal{Q}$ with quantization step $\delta>0$, pre-trained weights $\hat{w}\in\mathbb{R}^n$, the following analysis shows that **the $\ell_2$ quantization error is linear in $\delta$**:
> $$|| Xw_q - X\hat{w}|| \leq ||X \mathcal{Q}(\hat{w}) - X\hat{w}|| \leq \sigma_{\max}(X) ||\mathcal{Q}(\hat{w}) - \hat{w}|| \leq \frac{\sigma_{\max}(X) \sqrt{n}}{2}\delta.$$
> Our additional experiments also demonstrate that **the layer-wise quantization errors indeed are reduced by applying MagR on randomly sampled layers; see the figure in the attached pdf file**.
>
>   - We also note that MagR can preserve all the layers' outputs (before performing quantization), ensuring that the pre-trained model's performance remains unaffected. **This is crucial in the PTQ setting, as the goal is to search for the quantized model only within the local neighborhood of the pre-trained model**. The following table shows that MagR preprocessing indeed approximately maintains the perplexity (ppl) of the pre-trained model with minor degradation. Since we are minimizing the regularization $ \frac{1}{2}|| Xw - X\hat{w}||^2 + \alpha ||w||_{\infty}$ for preprocessing, for the minimizer $w^*$ it holds that $|| Xw^* - X\hat{w}|| = O(\sqrt{\alpha})\to 0$, as $\alpha \to 0$. When choosing a small $\alpha$ (equivalent to imposing large penalty on the fidelity term),  $|| Xw^* - X\hat{w}||$ will be small but not 0. This explains why the model performance degrades slightly after MagR (before quantization).
>
>     | Model   | Method | Wikitext2 (PPL)       | C4 (PPL) |
>     |----------|-|----------------|--------------|
>     | LlaMa2-7B    | Original | 5.47     | 6.97     |
>     |                       | After MagR | 5.52     | 7.04     |
>     | LlaMa2-13B    | Original | 4.88     | 6.46     |
>     |                         | After MagR | 4.92     | 6.52     |
>     | LlaMa2-70B    | Original | 3.31     | 5.52     |
>     |                         | After MagR | 3.35     | 5.56     |
>
> **Questions:**
>
> - **Ablation studies**:  $\alpha$ is the parameter balancing the tradeoff between the output discrepancy and the max magnitude of the weights. We show the ablation study on $\alpha$ for channel-wise quantization on LLaMA2-7B as below. Note that **the choice of $\alpha$ does not depend on the bit-width**, and $\alpha=0.001$ works well for all channel-wise quantization.
>
>   | Model   | $\alpha$ | W/A | Wiki (PPL)       | C4 (PPL) |
>   |----------|-|---------|-------|--------------|
>   | LlaMa2-7B    | 0.005 | 4/16 | 5.84     | 7.55     |
>   |                       | **0.001** | 4/16 | **5.70**     |  **7.28**    |
>   |                       | 0.0005 | 4/16 | 5.72     | 7.29     |
>   |                       | 0.0001 | 4/16 | 5.78     |  7.35     |
>   |                       | 0.00001 | 4/16 | 5.81     | 7.40     |
>   |     |     |       |          |
>   |                       | 0.005 | 3/16 | 6.64     | 8.74     |
>   |                       | **0.001** | 3/16 | **6.41**     |  **8.23**    |
>   |                       | 0.0005 | 3/16 | 6.49    | 8.38     |
>   |                       | 0.0001 | 3/16 | 6.83    |  8.79     |
>   |                       | 0.00001 | 3/16 | 7.08  | 9.19      |
>
>   | Model   | $\beta$ | W/A | Wiki (PPL)       | C4 (PPL) |
>   |----------|-|---------|-------|--------------|
>   | LlaMa2-7B    | 1    | 3/16 | 6.43     | 8.33     |
>   |                       | **0.9** | 3/16 | 6.41     | **8.23**     |
>   |                       | 0.85 | 3/16 | 6.48     |  8.39     |
>   |                       | 0.8 | 3/16 | 7.12     | 9.46     |
>   | | | | | |
>   |                       | 1      | 2/16 | 16.99     | 24.12     |
>   |                       | 0.9   | 2/16 | 20.88     |  31.78    |
>   |                       | 0.85 | 2/16 | 16.76     | 24.45     |
>   |                       | **0.8** | 2/16 | 16.73     |  **23.73**     |
>
> - **Why MagR+OPTQ not achieve better results in Table 3**:  The reasoning tasks in Table 3 require diverse knowledge and skills. Quantization may impede the model's ability to excel across all tasks. Therefore, a quantized model with better perplexity does not necessarily imply higher multi-task accuracy, especially when the perplexity values are close.
>
> - **Impact of outliers on PTQ**: Outliers have much larger magnitude than other weights, which result in a large quantization step and a large quantization error. MagR addresses this issue by minimizing $\ell_\infty$-norm to reduce the quantization step.

---

> > ### Comment · Reviewer_HDx9 · 2024-08-13
> >
> > Thank you for the response. After reading it and other reviewer's comments, I will raise my score.

---

> > > ### Author Response · Authors · 2024-08-14
> > > **Thanks for your reply**
> > >
> > > We sincerely appreciate your valuable feedback and thank you for raising your score!

---

### Official Review · Reviewer_AvoN · 2024-07-11

**Soundness:** 2
**Presentation:** 3
**Contribution:** 3
**Rating:** 5
**Confidence:** 4

**Summary:**

Authors propose an optimization-based preprocessing technique called MagR to enhance the performance of post-training quantization.
MagR adjusts weights by solving an l1-regularized optimization problem, reducing the maximum magnitude and smoothing out outliers.
As a nonlinear transformation, MagR eliminates the need for additional post-processing, thus avoiding overhead during inference.
Experiments demonstrate that MagR achieves state-of-the-art performance on the Llama family of models, such as significantly reduced perplexity on Wikitext2 for the LLaMA2-70B model.

**Strengths:**

1.The experiment results are good.
2. The idea is novel, which is using preprocessing technology before the quantization process.

**Weaknesses:**

In fact, I like the work but not in authors' view. I think this work is improve the generalization of model via norm. If this paper presents their work in this view, I would like to accept more.
1.Some value is lack of explanation, like " minimal the singular values of the feature matrix are less than 0.01 times the maximum singular value."
2. The applicability and the nature analysis of methods should be discussed.

**Questions:**

1. In Section 4.1, the singular values of the feature matrix are less than 0.01 times the maximum singular value. How is 0.01 calculated? If this value is changed, relaxing the constraint, will the results change?"
2. If the feature matrix is not actually rank-deficient, could the performance of the MagR method potentially suffer?
3. Proximal gradient descent may require more computational resources.
4. The model performance after preprocessing and before quantization should be presented. I wonder that the nature of this method is improve the generalizability of model before quantization.

**Limitations:**

The paper is lack of discussing the nature of preprocessing, i.e., what the preprocessing do for quantization.

---

> ### Author Rebuttal · Authors · 2024-08-02
>
> Thank you for the review! The primary concerns were regarding:
>
> **Questions:**
> - **Rank deficiency of feature matrix**: Section 4.1, specifically Table 1, demonstrates that the feature matrices across all the layers of the LLaMA family models have very small singular values, less than 0.01 times the maximum singular value. This indicates that the feature matrices are **approximately** rank-deficient, which ensures that we can approximately preserve the original model's performance after applying MagR. It is important to note that **exact** rank-deficiency is not a requirement for MagR to work. Notably, this approximate rank-deficiency is not unique to the LLaMA models; it also applies to the other models like OPT, as discussed in the QuIP paper. The threshold value of 0.01 is not a calculated parameter nor an algorithmic hyperparameter. Changing this threshold will not alter the feature matrices or affect the quantization performance of the proposed method.
>
> - **Efficiency of proximal gradient descent**: The preprocessing times on a single A100 GPU are 15 min for Llama2-7B, 30 min for 13B, and 3.5 hr for 70B. We believe that MagR can be readily applied to larger LLMs with 100+B parameters.
>
> - **Model performance after MagR**: The table at the bottom shows that the perplexity values degrade slightly after MagR. But since the range of the weights are reduced (Figure 1), we are able to use a smaller quantization step for the quantizer, which gives a smaller quantization error (see the table below) and improves the quantization performance.
>
> **Limitations**:
>
> - **Why MagR works:**
>   - MagR effectively reduces the range of the pre-trained weights by employing $\ell_\infty$-minimization, as illustrated by Figure 1. Given that the quantization step (or float scalar) is linearly proportional to the range of the pre-trained weights for a fixed bit-width (by the definition of uniform quantizer), **MagR results in a smaller quantization step. This, in turn, leads to a smaller quantization error**. In fact, given the activations $X\in\mathbb{R}^{m\times n}$, quantizer $\mathcal{Q}$ with quantization step $\delta>0$, pre-trained weights $\hat{w}\in\mathbb{R}^n$, the following analysis shows that **the $\ell_2$ quantization error is linear in $\delta$**:
> $$|| Xw_q - X\hat{w}|| \leq ||X \mathcal{Q}(\hat{w}) - X\hat{w}|| \leq \sigma_{\max}(X) ||\mathcal{Q}(\hat{w}) - \hat{w}|| \leq \frac{\sigma_{\max}(X) \sqrt{n}}{2}\delta.$$
> Our additional experiments also demonstrate that **the layer-wise quantization errors indeed are reduced by applying MagR on randomly sampled layers; see the figure in the attached pdf file or the table below**.
>
>     | Model (4-bit)   | Layer | RMSE With MagR      | RMSE Without MagR |
>     |----------|-|----------------|--------------|
>     | LlaMa2-7B    | 14 | 0.1327     | 0.1575     |
>     |                       | 65 | 0.1421     | 0.1616     |
>     |                       | 121 | 0.1622     | 0.1879     |
>     |                       | 184 | 0.2025     | 0.2319     |
>     |                       | 217 | 0.2198     | 0.2542     |
>     | LlaMa2-13B  | 14 | 0.1289     | 0.1518     |
>     |                       | 88 | 0.1647     | 0.1892     |
>     |                       | 145 | 0.1836     | 0.2127     |
>     |                       | 215 | 0.2098     | 0.2435     |
>     |                       | 271 | 0.2245     | 0.2618     |
>
>   - We also note that MagR can preserve all the layers' outputs (before performing quantization), ensuring that the pre-trained model's performance remains unaffected. **This is crucial in the PTQ setting, as the goal is to search for the quantized model only within the local neighborhood of the pre-trained model**. The following table shows that MagR preprocessing indeed approximately maintains the perplexity (ppl) of the pre-trained model with minor degradation. Since we are minimizing the regularization $ \frac{1}{2}|| Xw - X\hat{w}||^2 + \alpha ||w||_{\infty}$ for preprocessing, for the minimizer $w^*$ it holds that $|| Xw^* - X\hat{w}|| = O(\sqrt{\alpha})\to 0$, as $\alpha \to 0$. When choosing a small $\alpha$ (equivalent to imposing large penalty on the fidelity term),  $|| Xw^* - X\hat{w}||$ will be small but not 0. This explains why the model performance degrades slightly after MagR (before quantization).
>
>     | Model   | Method | Wikitext2 (PPL)       | C4 (PPL) |
>     |----------|-|----------------|--------------|
>     | LlaMa2-7B    | Original | 5.47     | 6.97     |
>     |                       | After MagR | 5.52     | 7.04     |
>     | LlaMa2-13B    | Original | 4.88     | 6.46     |
>     |                         | After MagR | 4.92     | 6.52     |
>     | LlaMa2-70B    | Original | 3.31     | 5.52     |
>     |                         | After MagR | 3.35     | 5.56     |

---

### Official Review · Reviewer_jujD · 2024-07-12

**Soundness:** 2
**Presentation:** 2
**Contribution:** 3
**Rating:** 7
**Confidence:** 3

**Summary:**

This paper introduces Weight Magnitude Reduction (MagR), a technique designed to smooth out outliers before LLM quantization. MagR adjusts pre-trained floating-point weights by solving an ℓ∞-regularized optimization problem. This preprocessing step reduces the maximum weight magnitudes, making the LLMs more suitable for quantization and resulting in better perplexity/accuracy results compared to models without preprocessing. Importantly, the MagR technique does not introduce any additional computational overhead during inference.

**Strengths:**

1) Since MagR is a technique for preprocessing LLM weights before quantization, it can be used in conjunction with other quantization methods (e.g. OPTQ).

2) MagR introduces proximal gradient descent steps, and it addresses these through layer-wise optimization using the Hessian, resulting in a shorter quantization runtime similar to OPTQ.

3) This method does not require additional operations for scaling activation, as needed in AWQ and QuIP, thereby avoiding extra overhead during inference.

**Weaknesses:**

1) MagR does not show significant improvement in perplexity/accuracy over previous method - QuIP.
Additionally, DecoupleQ [1] presents strong results for 2-bit quantization. While I understand that DecoupleQ was released recently, allowing insufficient time for a thorough comparison, I recommend the authors review DecoupleQ.

2) The paper emphasizes that MagR has no inference overhead, unlike AWQ and QuIP, which require scaling activations. However, it does not provide any latency information to highlight the significance of removing this additional inference overhead.

3) The runtime cost comparison between MagR and other methods is insufficient. The paper only provides the runtime of MagR, mentioning roughly that it is half the runtime of OmniQuant on page 8, lines 252-254. Since MagR is a preprocessing method, it is unclear if this comparison holds when considering the entire quantization process (MagR + OPTQ). Additionally, the runtime of QuIP is not addressed.

4) Some prior works have regularized weight distribution to make the model more suitable for quantization [2]. However, this paper does not mention or compare the differences between the proposed method and these earlier approaches.

5) The paper does not include any information about the calibration dataset.

[1] Guo, Yi, et al. "decoupleQ: Towards 2-bit Post-Training Uniform Quantization via decoupling Parameters into Integer and Floating Points." arXiv preprint arXiv:2404.12759 (2024).

[2] Kundu, Arnav, et al. "R2 Loss: Range Restriction Loss for Model Compression and Quantization." arXiv preprint arXiv:2303.08253 (2023).

**Questions:**

1) What is the inference latency of the proposed method compared to QuIP?

2) What is the runtime for QuIP and OmniQuant?

3) Why did you not provide perplexity results for the 7B/13B models for QuIP in Table 2?

4) Is QuIP incompatible with group-wise quantization, or can it be adapted to group-wise quantization with minor modifications, as done with MagR and OPTQ?

5)  For the results in Table 3, do all the methods use channel-wise quantization, or do some use group-wise quantization?

6) What type of calibration dataset did you use, and how many data points were included for calibration?

7) What happens if you apply Omniquant/QuIP after processing LLMs with MagR?

**Limitations:**

The proposed MagR is a promising approach, as it regularizes weight values to produce LLMs that are better suited for quantization. However, this paper lacks sufficient evaluation to convincingly demonstrate the superiority of the proposed method compared to state-of-the-art techniques.

---

> ### Author Rebuttal · Authors · 2024-08-03
>
> Thank you for the review and for pointing out relevant references. We'll add the references [1,2] and discussions.
>
> The primary concerns were regarding:
>
> **Weaknesses:**
>
> - **MagR vs QuIP, DecoupleQ**:
>
>   - It is possible to run additional coordinate descent iterations on top of OPTQ to further pursue the ppl, as shown in the updated results for the 2-bit 70B model. We also include the ppl results on 7B/13B models for QuIP. Our method outperforms QuIP in most cases. Regardless, QuIP trades off inference speed for accuracy, whereas MagR does not.
>
>     | Model   | Method | Wbits |Wiki (PPL)       | C4 (PPL) |
>     |----------|-|-|---------------|--------------|
>     | LlaMa2-7B    | QuIP               | 4 |5.94     | 8.01     |
>     |                       | MagR+OPTQ | 4 | 5.70    | 7.28     |
>     |                       | QuIP               | 3 |6.50     | 8.74     |
>     |                       | MagR+OPTQ | 3 | 6.41     | 8.23     |
>     |                       | QuIP               | 2 |27.13    | 31.33   |
>     |                       | MagR+OPTQ | 2 | 16.73    | 23.73  |
>     ||||||
>     | LlaMa2-13B  | QuIP             | 4 |5.01     | 6.88     |
>     |                       | MagR+OPTQ | 4 | 4.97    | 6.63     |
>     |                       | QuIP               | 3 |5.34     | 7.34     |
>     |                       | MagR+OPTQ | 3 | 5.41     | 7.19     |
>     |                       | QuIP               | 2 |10.09    | 13.13   |
>     |                       | MagR+OPTQ | 2 | 11.14     | 14.45     |
>     ||||||
>     | LlaMa2-70B    | QuIP               | 2 |6.33     | 8.94     |
>     |                         | MagR+OPTQ | 2 | 5.95    | 8.53   |
>
>   - DecoupleQ proposes to use block-wise optimization (similar to Omniquant) on top of GPTQ to refine the solution. Bottomline is that decoupleQ is still a quantization method (like GPTQ/Omniquant), MagR can be used as preprocessing for DecoupleQ to further improve its performance (MagR + DecoupleQ).
>
> - **Quantization runtime comparison**: We reported the runtime of MagR+OPTQ and MagR+RTN in Table 4. We meant that the entire runtime of MagR+OPTQ is roughly half of Omniquant. The following table shows the quantization runtimes of QuIP and OmniQuant. All the methods were tested on a single NVIDIA A100 GPU.
>
>   | Method | LlaMa2-7B       | LlaMa2-13B |
>   |-----------|---------------|--------------|
>   | QuIP               | 54 min     | 68 min     |
>   | Omniquant     |  73 min    | 2 hr     |
>   | OPTQ     |  22 min    | 40 min     |
>   | MagR+OPTQ  |  35 min     | 70 min    |
>
> - **Prior works on weight regularization**: Thanks for pointing out the reference. The goal of R2 Loss is similar to MagR,which regularizes a smaller range, but the reference uses a traditional end-to-end training. The training is from scratch and the regularization term is added to the original cross-entropy loss. The targets are CNN models. This method cannot be used for LLMs since the pre-training phase is too expensive to repeat. Moreover, their minimization is carried out by the conventional (sub)gradient method, whereas we investigate efficient algorithm which takes advantage of the proximal operator of $\ell_\infty$-norm.
>
> - **Calibration set**: The calibration dataset consists of 128 randomly selected 2048-token segments (context length) from WikiText2, which follows the routine of prior PTQ works.
>
> **Questions:**
> - **Inference efficiency**: MagR essentially replaces the original pre-trained weights with a new set of weights that have a smaller magnitudes prior to actual quantization, without sacrificing the original accuracy or altering the architecture. Consequently, our MagR+OPTQ achieves **exactly the same inference efficiency** as OPTQ. This is immediately supported by the widely-used inference kernel from the AutoGPTQ library. In contrast, at inference time, QuIP requires performing a random linear transformation on the activations before multiplying the quantized weights. Since the code for QuIP's inference is not released, we cannot compare them directly.  But according to QuIP's own report, QuIP's inference speed is at least **1.5 times slower than OPTQ** for the OPT-66B model (81 ms vs 53 ms), in addition to the power consumption overhead.
>
>
> - **Is QuIP incompatible with group-wise quantization?**: We think that QuIP should be compatible with group-wise quantization. QuIP first applies random linear transformations to preprocess the weights and activations and to smooth out outliers, then uses OPTQ to perform the actual quantization. Like OPTQ, it should be compatible with group-wise quantization. But they never reported such results. The drawback of QuIP  is that it requires random linear transformations on the feature matrix (so-called incoherence processing) not only in the preprocessing stage, but also in the inference phase.
>
> - **Results in Table 3**: All the results in Table 3 are for channel-wise quantization.
>
> - **Omniquant/QuIP with MagR**: We believe that MagR could enhance the performance of Omniquant or QuIP, but the improvement may not be as significant as with OPTQ. For instance, QuIP's incoherence processing also helps reduce weight magnitude, as reported in the paper. Omniquant uses block-wise minimization and learns the quantization step (weight clipping parameter) via SGD. In contrast, OPTQ is a fast, greedy, gradient-free algorithm. MagR+OPTQ achieves both simplicity and effectiveness, without introducing any inference overhead.

---

> ### Comment · Reviewer_jujD · 2024-08-10
>
> Thank you for your careful and detailed response.
>
> While I find the concept of MagR in regularizing weight values before quantization interesting, the paper seems to lack essential information needed to verify the significance of the proposed work:
>
> 1.	As you correctly noted, R2 Loss [2] was evaluated on CNNs using a full training approach. Since the loss term designed to regularize the weight distribution can be applied across various model architectures, it is crucial to thoroughly clarify the differences between R2 Loss and MagR, particularly in terms of loss term design or training efficiency. Although the difference between R2 Loss and MagR is briefly discussed in the rebuttal, the explanation provided is not detailed enough to fully clarify the distinction.
>
> 2.	The proposed MagR+OPTQ does not consistently achieve the best ppl/accuracy results compared to previous works.
>
> 3.	If the key novelty of MagR lies in pre-processing LLMs before quantization, it should be compatible with various quantization methods to enhance ppl/accuracy. However, the paper only discusses MagR+OPTQ without exploring the results of MagR combined with other quantization methods. Consequently, it is unclear whether MagR should be regarded as a general pre-processing solution or if ‘MagR+OPTQ’ is intended as a new quantization scheme. If MagR is indeed a broadly applicable pre-processing method, it would be highly valuable. If not, I have reservations about the significance of ‘MagR+OPTQ,’ given the limited ppl/accuracy improvement as discussed earlier.
>
> Therefore, I still perceive MagR as a limited contribution and will maintain my stance.

---

> ### Author Response · Authors · 2024-08-11
>
> ---
> 1. ... it is crucial to thoroughly clarify the differences between R2 Loss and MagR, particularly in terms of loss term design or training efficiency ...
> ---
> As we previously noted, R2 regularization applies $\ell_\infty$ penalty to the **traditional network loss** during end-to-end model pre-training, optimized using **standard SGD**, which is **practically infeasible for LLMs**. In contrast, MagR, as a concurrent approach, operates directly on the pre-trained model in a layer-by-layer manner, utilizing **linear least squares loss** to preserve each layer's output. MagR employs **proximal gradient descent** algorithm specifically tailored for this objective, **enabling efficient processing of LLMs**. This mathematically elegant algorithm represents a main contribution of our work.
>
> ---
> 2. ... MagR+OPTQ does not consistently achieve the best ppl/accuracy results ...
> ---
>
> While we don’t claim that MagR+OPTQ always achieves the highest accuracy, it does perform at or near the top in most cases. Moreover, its accuracy can be further enhanced by applying techniques like additional coordinate descent iterations or learnable weight clipping as suggested by Reviewer 5GSj. Accuracy is just one of many critical performance metrics. More importantly, we introduce a technique that **incurs no additional overhead at inference time while achieving both training efficiency and accuracy comparable to the state-of-the-art, which typically sacrifices inference speed**. Inference speed is crucial for real-world applications, particularly in resource-limited settings such as edge computing.
>
> ---
> 3. ... the paper only discusses MagR+OPTQ without exploring the results of MagR combined with other quantization methods ...
> ---
>
>  We would like to remind the reviewer that **we also reported the substantial performance gain of MagR over the nearest round method (MagR+RTN vs RTN) in Table 2.**
>
> |Method | Wbits |Wiki (PPL)| | || C4 (PPL)|||
> |-|-|-|-|-|-|-|-|-|
> || |7B|13B|70B|-|7B|13B|70B|
> |Baseline| FP16|5.47|4.88|3.31||6.97|6.46|5.52|
> ||||||
> |RTN|4/16|6.11|5.20|3.67|-|7.71|6.83|5.79|
> |MagR+RTN|4/16|5.91|5.17|3.58|-|7.52|6.81|5.72|
> ||||||
> |RTN|3/16|539.48|10.68|7.52|-|402.35|12.51|10.02|
> |MagR+RTN|3/16|8.66|6.55|4.64|-|10.78|8.26|6.77|
>
> Here we also demonstrate that MagR can enhance the performance of QuIP:
>
> |Model|Method|Wbits|Wiki (PPL)| C4 (PPL)|
> |-|-|-|-|-|
> |LlaMa2-7B|QuIP|4|5.94|8.01|
> | |MagR+QuIP|4|5.74|7.25|
> | |QuIP|3|6.50|8.74|
> | |MagR+QuIP|3|6.25|7.88|
> | |QuIP|2|27.13|31.33|
> | |MagR+QuIP|2|13.31|14.49|
> ||||||
> |LlaMa2-13B|QuIP|4|5.01|6.88|
> | |MagR+QuIP|4|4.99|6.63|
> | |QuIP|3|5.34|7.34|
> | |MagR+QuIP|3|5.29|7.02|
> | |QuIP|2|10.09|13.13|
> | |MagR+QuIP|2|9.40|11.07|
>
>
> In summary, MagR preserves the behavior of the pre-trained model, as demonstrated by the new data, while allowing for small quantization steps in the subsequent PTQ process, leading to reduced quantization error. So theoretically, MagR can be combined with other PTQ methods.

---

> > ### Comment · Reviewer_jujD · 2024-08-12
> >
> > Thank you for your response.
> >
> > After reading your reply, I have decided to raise my score to 7, as the additional experimental results on MagR combined with other quantization methods have convinced me of the effectiveness of the proposed work.
> >
> > However, I would like to request that the authors properly cite previous works related to weight regularization. While I understand that this paper is not identical to earlier approaches, applying weight regularization to achieve quantization-friendly models is a well-studied technique, and this paper currently lacks citations of these relevant works.

---

> > > ### Author Response · Authors · 2024-08-12
> > >
> > > Thank you for your feedback and for raising your score. We appreciate your recognition of the contributions of our work. We will thoroughly review and cite previous works related to weight regularization that are pertinent to our approach.

---

### Author Rebuttal · Authors · 2024-07-31

We thank the reviewers for their constructive feedback. We address the common concerns below:

**(1) Why MagR works:**
- MagR effectively reduces the range of the pre-trained weights by employing $\ell_\infty$-minimization, as illustrated by Figure 1. Given that the quantization step (or float scalar) is linearly proportional to the range of the pre-trained weights for a fixed bit-width (by the definition of uniform quantizer), **MagR results in a smaller quantization step. This, in turn, leads to a smaller quantization error**. In fact, given the activations $X\in\mathbb{R}^{m\times n}$, quantizer $\mathcal{Q}$ with quantization step $\delta>0$, pre-trained weights $\hat{w}\in\mathbb{R}^n$, the following analysis shows that **the $\ell_2$ quantization error is linear in $\delta$**:
$$|| Xw_q - X\hat{w}|| \leq ||X \mathcal{Q}(\hat{w}) - X\hat{w}|| \leq \sigma_{\max}(X) ||\mathcal{Q}(\hat{w}) - \hat{w}|| \leq \frac{\sigma_{\max}(X) \sqrt{n}}{2}\delta.$$
Our additional experiments also demonstrate that **the layer-wise quantization errors indeed are reduced by applying MagR on randomly sampled layers; see the figure in the pdf file**.

- We also note that MagR can preserve all the layers' outputs (before performing quantization), ensuring that the pre-trained model's performance remains unaffected. **This is crucial in the PTQ setting, as the goal is to search for the quantized model only within the local neighborhood of the pre-trained model**. The following table shows that MagR preprocessing indeed approximately maintains the perplexity (ppl) of the pre-trained model with minor degradation. Since we are minimizing the regularization $ \frac{1}{2}|| Xw - X\hat{w}||^2 + \alpha ||w||_{\infty}$ for preprocessing, for the minimizer $w^*$ it holds that $|| Xw^* - X\hat{w}|| = O(\sqrt{\alpha})\to 0$, as $\alpha \to 0$. When choosing a small $\alpha$ (equivalent to imposing large penalty on the fidelity term),  $|| Xw^* - X\hat{w}||$ will be small but not 0. This explains why the model performance degrades slightly after MagR (before quantization).

  | Model   | Method | Wikitext2 (PPL)       | C4 (PPL) |
  |----------|-|----------------|--------------|
  | LlaMa2-7B    | Original | 5.47     | 6.97     |
  |                       | After MagR | 5.52     | 7.04     |
  | LlaMa2-13B    | Original | 4.88     | 6.46     |
  |                         | After MagR | 4.92     | 6.52     |
  | LlaMa2-70B    | Original | 3.31     | 5.52     |
  |                         | After MagR | 3.35     | 5.56     |

**(2) Inference efficiency:** MagR essentially replaces the original pre-trained weights with a new set of weights that have a smaller magnitudes prior to actual quantization, without sacrificing the original accuracy or altering the architecture. Consequently, our MagR+OPTQ achieves **exactly the same inference efficiency** as OPTQ. This is immediately supported by the widely-used inference kernel from the AutoGPTQ library. In contrast, at inference time, QuIP requires performing a random linear transformation on the activations before multiplying the quantized weights. Since the code for QuIP's inference is not released, we cannot compare them directly.  But according to QuIP's own report, QuIP's inference speed is at least **1.5 times slower than OPTQ** for the OPT-66B model (81 ms vs 53 ms), in addition to the power consumption overhead.

**(3) Preprocessing time:**  We would like to highlight that a major contribution of our work is efficiently addressing the computational challenge of large-scale $\ell_\infty$-regularization with matrix variables. The preprocessing times for MagR (without quantization) on **a single A100 GPU** are modest: **15 min for Llama2-7B, 30 min for 13B, and 3.5 hr for 70B**. We believe that MagR is readily applicable to larger LLMs with 100+B parameters.

**(4) Ablation study on** $\alpha$:  $\alpha$ is the parameter balancing the tradeoff between the output discrepancy and the max magnitude of the weights. We show the ablation study on $\alpha$ for channel-wise quantization on LLaMA2-7B as below. Note that **the choice of $\alpha$ does not depend on the bit-width** and $\alpha=0.001$ is the best choice for channel-wise quantization.

  | Model   | $\alpha$ | W/A | Wiki (PPL)       | C4 (PPL) |
  |----------|-|---------|-------|--------------|
  | LlaMa2-7B    | 0.005 | 4/16 | 5.84     | 7.55     |
  |                       | **0.001** | 4/16 | **5.70**   |  **7.28**    |
  |                       | 0.0005 | 4/16 | 5.72     | 7.29     |
  |                       | 0.0001 | 4/16 | 5.78     |  7.35     |
  |                       | 0.00001 | 4/16 | 5.81     | 7.40     |
  |     |     |       |          |
  |                       | 0.005 | 3/16 | 6.64     | 8.74     |
  |                       | **0.001** | 3/16 | **6.41**     |  **8.23**    |
  |                       | 0.0005 | 3/16 | 6.49    | 8.38     |
  |                       | 0.0001 | 3/16 | 6.83    |  8.79     |
  |                       | 0.00001 | 3/16 | 7.08  | 9.19      |



In light of our responses to the reviewers' concerns, we would be very grateful if you would reconsider your opinion. We believe our work proposes a simplistic and effective method for quantizing LLMs without introducing any inference overhead.

---

> ### Comment · Reviewer_5GSj · 2024-08-09
> **QuIP quantization error**
>
> Dear authors,
>
> Thank you for your thorough response to our comments and questions. I have one more question.
>
> In the attached figure, you compare MagR's quantization error with OPTQ. Do you have time to add the quantization error with the transformed weights of QuIP? This would be quite useful in understanding whether your optimization is theoretically superior to the random linear transformation of QuIP (beyond QuIP's additional compute overhead).
>
> Also, is the quantization error calculated before or clipping, namely, is $\beta$<1 for these results?
>
> Thanks

---

> ### Author Response · Authors · 2024-08-10
> **QuIP Minimizes a Different Quantization Error**
>
> Thank you for your questions.
>
> In response to the second question, we evaluated the quantization errors for INT4 quantization, and we did not apply weight clipping at 4-bit (i.e., $\beta=1$). This means that the reduction in quantization error is entirely due to the MagR preprocessing.
>
> Regarding the quantization error with QuIP, it targets a different quantization error in a distinct parameter space compared to our method or OPTQ. Specifically, we solve $\min_{W_q\in\mathbb{Q}} ||X(W_q - W)||^2$, where the focus is directly on minimizing the error in the original weight space. In contrast, QuIP first applies orthogonal transformations $U$ and $V$ to the weights and activations before quantization: $\min_{W_q} ||XV V^{\top} (W_q - W)U||^2 := \min_{\tilde{W_q}\in\mathbb{Q}} ||\tilde{X} (\tilde{W_q} - \tilde{W})||^2$. Here, $\tilde{X} = XV$, $\tilde{W_q} = V^{\top} W_q U$, and $\tilde{W} = V^{\top} W U$ represent the transformed activations and weights. QuIP only quantizes the transformed weights $\tilde{W_q}$, and their corresponding $W_q$ is actually float. In contrast, we directly quantize $W_q$ as is done in standard OPTQ. As a result, the quantization errors between our method and QuIP are not directly comparable.
>
> We hope we have addressed the reviewer’s questions.

---

> ### Author Response · Authors · 2024-08-12
>
> We hope our rebuttal have addressed the reviewers' concerns, and we would greatly appreciate it if you could reconsider your rating.

---

### Decision · Program_Chairs · 2024-09-25

**Decision:**

Accept (poster)

**Comment:**

This paper studies a post-training quantization (PTQ) method to compress LLMs, such as the LLAMA family of models. Recent PTQ approaches use linear transformations to make quantization easier but introduce computational overhead when making predictions. Therefore, this paper proposes a new PTQ method, MagR, that does not increase the inference computation. The main idea is to optimize the pretrained weights to have a smaller l-inf norm without changing the original layer-wise features, which is formulated in Eqn. (2). When the range of pre-trained weights is reduced, the quantization step (which is proportional to the range) is therefore reduced, which leads to reduced quantization error. The computation of the l-inf constrained optimization problem is solved efficiently using a proximal gradient descent algorithm. The paper shows promising results in the sub-4bit quantization regime (INT4/INT3/INT2) when compared to other PTQ methods.
Reasons for acceptance:
- Novel idea in reducing the weight range for PTQ.
- Fast calculation of the L-inf regularization problem (15 min for Llama2-7B on a single A100 GPU).
- No overhead in inference time compared to other linear-transform-based PTQ methods.
- Principled methods for weight reduction and a clear explanation of why the idea of reducing the weight range works.
- Potential to be combined with other PTQ methods.

This paper achieved unanimous support from the reviewers after lots of discussions. The paper can be further improved after incorporating some of the discussions with the reviewers, e.g., explicit discussions on inference time savings, prior work on regularization for better model quantization, and other suggested experiments and ablation studies by the reviewers.